

# Atmospheric ammonia variability and link with PM formation: a case study over the Paris area

Viatte Camille[1], Wang Tianze[1], Van Damme Martin[2], Dammers Enrico [3], Meleux Frederik[4], Clarisse Lieven[2], Shephard Mark W.[3], Whitburn Simon[2], Coheur Pierre François[2], Cady-Pereira Karen E.[5], and Clerbaux Cathy[1,2]

[1] LATMOS/IPSL, Sorbonne Université, UVSQ, CNRS, Paris, France
[2] Université libre de Bruxelles (ULB), Service de Chimie Quantique et Photophysique, Atmospheric Spectroscopy, Brussels, Belgium
[3] Environment and Climate Change Canada, Toronto, Ontario, Canada
[4] Institut national de l'environnement industriel et des risques, INERIS, Verneuil en Halatte, France
[5] Atmospheric and Environmental Research (AER), Inc., Lexington, USA





## Abstract

The Paris megacity experiences frequent particulate matter ($PM_{2.5}$, PM with a diameter less than 2.5 μm) pollution episodes in springtime (March-April). At this time of the year, large parts of the particles consist of ammonium sulfate and nitrate which are formed from ammonia ($NH_3$) released during fertilizer spreading practices and transported from the surrounding areas to Paris. There is still limited knowledge on the emission sources around Paris, their magnitude and seasonality.

Using space-borne $NH_3$ observation records of 10-years (2008-2017) and 5-years (2013-2017) provided by the Infrared Atmospheric Sounding Interferometer (IASI) and the Cross-Track Infrared Sounder (CrIS) instrument, regional pattern of $NH_3$ variabilities (seasonal and inter-annual) are derived. Observations reveal identical high seasonal variabilities with three major $NH_3$ hot spots found from March to August. The high inter-annual variability is discussed with respect to atmospheric total precipitation and temperature.

A detailed analysis of the seasonal cycle is performed using both IASI and the CrIS instrument data, together with outputs from the CHIMERE atmospheric model. For months of high $NH_3$ concentrations (March to August) the CHIMERE model shows good correspondence with correlation slopes of 0.98 and 0.71 when comparing with IASI and CrIS, respectively. It is found that the model is only able to reproduce half of the observed atmospheric temporal $NH_3$ variability in the domain. In term of spatial variability, the CHIMERE monthly $NH_3$ concentrations in springtime show a slight underrepresentation over Belgium and the United-Kingdom and overrepresentation in agricultural areas in the French Brittany/Pays de la Loire and Plateau du Jura region, as well as in the north part of Switzerland.

Using HYSPLIT cluster analysis of back-trajectories, we show that $NH_3$ total columns measured in spring over Paris are enhanced when air masses are originated from the Northeast (e. g., Netherlands and Belgium), highlighting the long-range transport importance on the $NH_3$ budget over Paris.

Finally, we quantify the key meteorological parameters driving the specific conditions important for the $PM_{2.5}$ formation from $NH_3$ in the Ile-de-France region in springtime. Data-driven results based on surface $PM_{2.5}$ measurements from the Airparif network and IASI $NH_3$ observations show that a combination of the factors, e. g. a low boundary layer of ~500m, a relatively low temperature of 5°C and a high relative humidity of 70%, contributes to favor $PM_{2.5}$ and $NH_3$ correlation.



## 1. Introduction


Ammonia (NH$_3$) is an atmospheric pollutant and one of the main sources of reactive nitrogen in
the atmosphere which is involved in numerous biochemical exchanges impacting all ecosystems
[Sutton et al., 2013]. The global budget of reactive N has dramatically increased since the
preindustrial era [Holland et al., 2005; Battye et al., 2017] causing major environmental
damages such as ecosystems and species extinction, as well as soil and water eutrophication
and acidification [Rockström et al., 2009]. NH$_3$ is a precursor of ammonium salts which can form
up to 50% to particulate matter (PM) total mass [Behera et al., 2013]. Large cities such as Paris
(which is the most populated area in the European Union with 10.5 million people when its
larger metropolitan regions are included) typically experiences strong PM pollution episodes in
springtime. These particles are known to be harmful for human health [Pope III et al., 2009]
inducing 2000 deaths per year in the Paris megacity [Corso et al., 2016] and to impact the
radiative budget of the Earth [Myhre et al., 2013].
Because of their impact on the environment, public health, and climate change, NH$_3$ emissions
are regulated in several countries in the world. However, NH$_3$ emissions of European countries
have increased by 2% over the period 2014-2016 [National Emission Ceilings Directive reporting
status, 2018], where the Gothenburg Protocol set a reduction of 6% by 2020. In France, where
94% of NH$_3$ emissions come from the agriculture sector [CITEPA, 2018] as a result of extensive
fertilizer use to increase crop yields [Erisman et al., 2008], policies have been implemented with
the aim to reduce NH$_3$ emissions by 13% in 2030 related to 2005 [CEIP, 2016]. However NH$_3$
emissions are projected to increase in the future globally with increased population and food
demand [van Vuuren et al., 2011] and NH$_3$ volatilization will be enhanced with climate change
[Sutton et al., 2013].
Once in the atmosphere, NH$_3$ is rapidly removed by wet and dry deposition, and by reactions
with atmospheric sulfuric and nitric acid, leading to a relatively short lifetime between a few
hours and few days [Galloway et al., 2003]. Release of NH$_3$ in the atmosphere depends on i)
agriculture practices: spreading season, fertilizer form (urea, ammonium nitrate), fertilizer
application methods, crops, soil conditions such as pH [Hamaoui-Laguel et al., 2014]; and on ii)
meteorological conditions (i.e. wind, temperature, and precipitation). Inter-annual variability of
PM formation over urban area is poorly understood, since it also depends on many factors such
as atmospheric humidity and temperature, which govern the phase equilibrium of secondary
aerosols [Fuzzi et al., 2015]. The variety of factors influencing NH$_3$ volatilization and PM
formation illustrates the complexity of predicting their concentrations in the atmosphere
[Behera et al., 2013].
Atmospheric chemical transport models have difficulty representing both NH$_3$ and PM$_{2.5}$
distributions due to the challenge of reproducing NH$_3$ temporal variability [Pinder et al., 2006;





Fortems-Cheiney et al., 2016], long-range transport of pollutants [Moran et al., 2014], and
secondary aerosol formation in the atmosphere [Petetin et al., 2016]. The GEOS-Chem chemical
transport model [Bey et al., 2001] was found to underestimate the observed NH$_3$ concentrations
in most regions of the globe [Zhu et al., 2013; Li et al., 2017]. Heald et al. (2012) compared the
IASI observations with the GEOS-Chem model and showed that NH$_3$ is likely underestimated in
California, leading to a local underestimate of ammonium nitrate aerosol. Similarly, the French
CHIMERE model [Menut et al., 2013] underestimates the NH$_3$ budget over Paris [Petetin et al.,
2016; Fortems-Cheiney et al., 2016] because of the mis-representation of agricultural emissions
in terms of intensity and both spatial and temporal distribution. Often ground and aircraft-
based observations are used to provide detailed representation of the atmospheric state that
can be used to evaluate and improve the model simulations; however, these can be spatially
sparse and/or over short sampling periods, especially globally. Additionally, more recently
available (within the last 10-years) sun-synchronous satellite-based infrared sensors have been
providing NH$_3$ observations globally with a spatial resolution of ~15 km approximately twice a
day. These satellite observations have limited independent vertical information, but do capture
the spatiotemporal variabilities needed to help address these issues and improve model
simulations, especially in remote locations [Skjøth et al., 2011; Kranenburg et al., 2016].
Aside from the Tropospheric Emission Spectrometer (TES, [Beer et al., 2008]), now
decommissioned but which was first to demonstrate the capability of thermal infrared
instruments to monitoring lower tropospheric NH$_3$ , 3 missions are able to measure it now : the
Atmospheric InfraRed Sounder (AIRS, [Warner et al., 2016]*), the Cross-track Infrared Sounder
(CrIS, [Shephard and Cady-Pereira, 2015]), and the Infrared Atmospheric Sounding
Interferometer (IASI, [Clarisse et al., 2009]). Recent studies have shown the increased capacity
of space-borne instruments to derived spatial and seasonal distributions of NH$_3$ concentrations
globally [Clarisse et al., 2009; Shephard et al., 2011; Van Damme et al., 2014a & 2015a],
regionally [Beer et al., 2008; Clarisse et al., 2010; Van Damme et al., 2014b] and locally [Van
Damme et al., 2018], as well as trends of NH$_3$ [Warner et al., 2017].
Representative measurements of NH$_3$ concentrations and spatiotemporal variabilities are
needed to address the link between NH$_3$ and PM$_{2.5}$ formation and improve model simulations.
This has been attempted previously in some cities around the world, such as in Shanghai [Ye et
al., 2011], Houston [Gong et al., 2013], Santiago City [Toro et al., 2014], and Beijing [Zhao et al.,
2016] for instance. However, although the Paris megacity is repeatedly shrouded by particulate
pollution episodes, many of studies are limited and performed over relatively short time frame
during field campaigns [Petetin et al., 2016; Zhang et al., 2013], or based on numerical
simulations [Skyllakou et al., 2014]. Our study is a data-driven regional approach and considers a
longer time period to study the seasonal/inter-annual variabilities of NH$_3$ and its impact of PM$_{2.5}$
formation over the Paris megacity. Specifically in this paper we study concentrations and





spatiotemporal variability of atmospheric $NH_3$ from the agricultural sector to gain insights on its
effects on megacity air quality using: 1) long-term satellite observations derived from IASI (10
years from 2008 to 2017) and CrIS (5 years from 2013 to 2017) at regional scale (400km radius-
circle from Paris city center); 2) spatiotemporal patterns of the CHIMERE model evaluated
against the IASI and CrIS datasets for 2014 and 2015; and 3) the main meteorological
parameters favoring the secondary $PM_{2.5}$ formation from $NH_3$ in the Paris megacity are
analyzed.

## 124    2. Methodology

### 125    2.1.    Region of analysis

The domain of analysis covers a circular area of 400 km radius around the Paris city center
(Figure 1, larger circle) enabling the study of temporal and spatial variabilities of $NH_3$ emission
sources likely to affect air quality in the Paris megacity. It has been selected for two reasons.
First, it includes main regions known for their high $NH_3$ emissions, which can be transported and
affect air quality over the Parisian region (Ile-de-France –IdF-, smaller circle in Figure 1).
Emission regions in the Netherlands, North of Germany, Northwest of Belgium, and the Brittany
region in France, are highlighted in darker colors in Figure 1 (emissions values are from the
European Monitoring and Evaluation Programme -EMEP- 2015). Second, this area corresponds
to the transport of 24 hours back-trajectories from Paris generated from the HYSPLIT model for
one year, ensuring that $NH_3$ can indeed be efficiently transported from the emitting sources
within the selected domain to the IdF region.

### 137    2.2.    Satellite observations of ammonia

For this study we used the available date from IASI and CrIS which are both Fourier transform
spectrometers to evaluate the current capacity to observe $NH_3$ concentrations from space, and
study its variability around IdF. Technical information are summarized in Table 1.

### 141    2.2.1. Infrared Atmospheric Sounding Interferometer (IASI)

IASI is a nadir-viewing spectrometer launched on board the Metop-A and Metop-B satellites and
operated by EUMETSAT (European Organisation for the Exploitation of Meteorological
Satellites), since October 2006 and September 2012, respectively. These satellites are on similar
polar orbits with Equator crossing times at 09:30 (21:30) local mean solar time for the
descending (ascending) orbit. IASI measures the thermal infrared radiation of the system Earth-
atmosphere in the spectral range from 645 to 2760 $cm^{-1}$ with a spectral resolution 0.5 $cm^{-1}$
apodized. The satellite swath is an area of 2200 km width composed by off-nadir measurements
up to 48.3$^{\circ}$ on both sides of the track. At nadir, the IASI field of view is composed of 4 x 4 pixels
of 12 km diameter each [Clerbaux et al., 2009].



The NH$_3$ total columns used here are derived from IASI using an Artificial Neural Network
reanalyzed with ERA-interim data (ANNI-NH3-v2.1R [Van Damme et al., 2017]). This dataset is
consistent in time and suitable for investigating inter-annual variability, which is one purpose of
this study. Note that we have considered here only morning measurements (9:30) since the
evening ones (21:30) are associated with larger relative errors [Van Damme et al., 2017]. IASI
retrievals provide a robust error estimate for each IASI-NH3 observations, allowing to take into
account the variable sensitivity when comparing IASI dataset with independent measurements.
Finally, no filter on relative errors of the IASI datasets has been applied following
recommendations from Van Damme et al. (2017) and outliers for which concentrations exceed
10 standard deviations above the mean in the domain of study have been removed.
Over the studied area, Metop-A and Metop-B have an overpass time difference ranging from
only a few seconds to 67 minutes depending on the viewing geometry of the satellite scans; the
average difference is 26 minutes for the 1325 days of common measurements. Monthly maps
for the 10 years of observations between 2008 and 2017 are obtained by averaging Metop-A
and whenever Metop-B (the two instruments are considered jointly for their period of common
operation from March 2013 to 2017) with more than $10^5$ pixels on average over the domain of
analysis. The number of available NH$_3$ columns depends not only on the satellite overpass time
but also on the state of the atmosphere being remotely sensed (e.g. thermal contrast and cloud
cover). IASI NH$_3$ has been evaluated using the LOTOS-EUROS model over Europe [Van Damme et
al., 2014b] and ground-based and airborne measurements [Van Damme et al., 2015b], showing
consistency between the IASI NH$_3$ and the available datasets. When comparing IASI NH$_3$
(previous IASI-NN version) with ground-based Fourier transform infrared (FTIR) observations, a
correlation of 0.8 and a slope of 0.73, with a mean relative difference of −32.4 ± (56.3)% have
been found [Dammers et al., 2016].

### 175    2.2.2. Cross-track Infrared Sounder (CrIS)

The CrIS instrument [Zavyalov et al., 2013] is a Fourier Transform spectrometer operated by the
Joint Polar Satellite System (JPSS) program on Suomi National Polar-orbiting Partnership (NPP)
satellite, launched on 28 October 2011. CrIS is in a sun-synchronous orbit with a mean local
daytime overpass time of 13:30 (01:30) in the ascending (descending) node. CrIS measures the
atmospheric composition over three wavelength bands in the infrared region (645–1095 cm$^{-1}$;
1210–1750 cm$^{-1}$; 2155–2550 cm$^{-1}$). NH$_3$ retrievals are performed from the 645–1095 cm$^{-1}$ band
with a spectral resolution of 0.625 cm$^{-1}$. The CrIS instrument scans a 2200 km swath width (+/-
50 °). At nadir, the CrIS field of view consists of a 3 × 3 array of circular pixels of 14 km diameter
each.
The CrIS Fast Physical Retrieval (CRPR) [Shephard and Cady-Pereira., 2015] uses an optimal
estimation approach [Rodgers, 2000] that minimizes the difference between the CrIS measured



atmospheric spectra and a very fast Optimal Spectral Sampling (OSS) [Moncet et al., 2008]
forward model simulated spectrum to retrieve atmospheric profiles of ammonia volume mixing
ratios. This physical approach provides direct estimates of the retrieval errors and the vertical
sensitivity (averaging kernels) of the satellite observations, which is important as they vary from
profile-to-profile depending on the atmospheric state. The retrieved error covariance and
averaging kernels are also beneficial for air quality model comparisons and data assimilation
into models as any *a priori* information used in the retrieval can be accounted for in a robust
manner (i.e. observation operator). CrIS has been shown to retrieve ammonia surface
concentrations values down to ~0.2-0.3 ppbv under favorable conditions [Kharol, et al., 2018].
CrIS comparisons with ground-based FTIR observations show a correlation of 0.77 with a low
CrIS bias of +2% in the total column [Dammers et al., 2017]. Initial evaluation against surface
observations from the Ammonia Monitoring Network (AMoN) show that even with the inherent
sampling differences between the two surface observations they compare well with a
correlation of 0.76 and an overall mean CrIS – AMoN difference of ~+15% [Kharol et al., 2018].
For this study, the CrIS quality flag = 4 has been used, ensuring that retrievals provide some
information from the measurement (degrees-of-freedom- of-signal > 0.1). In addition, outliers
for which concentrations exceed 10 standard deviations above the mean have been removed.
## 2.3.    Modelling NH$_3$ from the CHIMERE model
The CHIMERE runs used in this study were obtained in the framework of the Copernicus
Atmospheric Monitoring Service (CAMS, https://atmosphere.copernicus.eu/), and its annual
task devoted to the production of regional reanalysis over Europe. The hindcasts for year 2014
and 2015 (raw simulation without data assimilation) were produced over Europe with a
horizontal         resolution         of         0.1°         per         0.1°         and
9 vertical levels stretched from the surface up to 500 hPa (~5000m). The input data to feed
CHIMERE [Menut et al., 2013; Mailler et al., 2017] were the Integrated Forecasting System (IFS)
meteorological data from European Centre for Medium-Range Weather Forecasts (ECMWF), the
annual emission inventory provided by the Netherlands Organisation for Applied Scientific
Research (TNO) [Kuenen et al., 2014] for year 2011 and the fire emissions from the Global Fire
Assimilation System (GFAS, [Kaiser et al., 2012]).The model computes hourly concentrations for
more than 180 species, among which are the regulated pollutants such as ozone, PM$_{10}$, and
NH$_3$. Within CHIMERE a comprehensive modelling system allows to compute the evolutions of
gaseous species and aerosols taking into account physical and chemical process. More than 30
gaseous species are involved in the chemical scheme and an aerosol module assesses the gas-
particulate phase equilibrium and compute the aerosol composition (inorganic, organic and
natural components).These datasets were evaluated over Europe for several pollutants before
being used for air quality studies (http://policy.atmosphere.copernicus.eu/Reports.html).



The model NH$_3$ profiles were integrated vertically along the 9 km model layers to provide a
column that can be compared to that of the satellite measurements. Concretely this makes the
reasonable assumption that all the NH$_3$ is located within this 0-5km layer (see e.g. Figure 1 in
[Whitburn et al., 2016]).

### 2.4. Relative scales and coincidence criteria for dataset comparisons

Direct quantitative comparisons of satellite NH$_3$ products are difficult because of the different
overpass times and ground footprint sizes of the 2 space borne instruments, which are not
compatible with the high variability of NH$_3$ in space and time. Therefore, the evaluation of
satellite observations is often made with the use of in situ measurements performed at surface
and onboard aircrafts [Nowak et al., 2012; Van Damme et al., 2015b], or with ground-based
remote-sounding FTIR [Dammers et al., 2016; Dammers et al., 2017].
The purpose here of comparing CrIS and IASI is to assess qualitatively the spatiotemporal
patterns of the NH$_3$ sources derived from the two datasets and use these regional observations
to evaluate the CHIMERE model in the domain of analysis at the local time for their respective
overpasses: 9:30 and 13:30. CHIMERE outputs, in terms of NH$_3$ concentrations, have already
been compared to the IASI observations at regional scale (Europe, [Fortems-Cheiney et al.,
2016], and to surface measurements at local scale (Paris, [Petetin et al., 2016]), but have never
been evaluated against the CrIS observations.
One aspect that needs to be considered when comparing concentration amounts inferred from
infrared satellite observations is the importance of the algorithm and the a priori information
used in the retrieval, especially for NH$_3$ which has limited vertical information. Some differences
between the IASI and CrIS observations might arise due to instrument measurement differences
(e.g. sensitivity), difference sampling period (e.g. overpass times of morning/evening vs middle
of day/night), and retrieval algorithm differences, but they have both been validated and shown
to capture well the spatiotemporal variations in lower tropospheric ammonia. Since the purpose
of our study is not to quantitatively compare IASI and CrIS NH$_3$ data, but rather to use these
independent datasets to assess NH$_3$ sources patterns over the domain and qualitatively
evaluate the CHIMERE model in term of NH$_3$ concentrations and variabilities, a standardization
procedure was applied to their retrieved absolute NH$_3$ columns. We computed "standardized
columns" for each independent dataset (IASI, CrIS, and CHIMERE, separately) for 2014 and 2015
over the domain of study in such a way that the corresponding values have a standard deviation
of 1 and a mean of 0, as in [Wilks, 2011].
In addition, to compare CHIMERE outputs with satellite data/columns, spatial and temporal
coincidence criteria have been applied. To compare satellite observations, all CrIS pixels located
within a 25-km radius circle from the center of the IASI ground pixels have been considered



within the same day of measurements. A spatial criterion of 25 km has been chosen because it
optimizes the number of pairs involved in the statistics and improves the correlations. As for the
comparisons between the model and the observations: all CHIMERE outputs located within the
same 0.15°x0.15° grid box than the satellite and within 1 hour from its measurement have been
selected.

## 3. Results

### 3.1. NH$_3$ regional observations derived from IASI (10-years) and CrIS (5-years)

#### 3.1.1. Seasonal variabilities

First the seasonal variability was investigated over the IdF area. On a monthly basis, the 10-year
and 5-year averaged regional NH$_3$ total column distributions derived from IASI and CrIS were
found to exhibit a high seasonality over the domain (Figures 2 and 3). Note that the distributions
in Figures 2 and 3 have been obtained by averaging satellite NH$_3$ observations in 0.25°x 0.25°
grid boxes. Both satellite datasets exhibits the same variability over the domain even if the time
period is different (10-years versus 5-years) and the sampling hour differs (~9.30 versus ~13.30).
One note that CrIS and IASI NH$_3$ columns present small differences in term of NH$_3$ total columns
in low concentration regimes in the domain of study.
In these figures (2 and 3) high NH$_3$ concentrations (up to $2.10^{16}$ molecules/cm$^2$) can be observed
from March to August at different locations of the domain:

- The French Champagne-Ardennes region in March and April (Figures 2 and 3, box A),
- The northern part of the domain corresponding to the Netherlands and the North of Belgium from April to August (Figures 2 and 3, box B), and
- The Brittany/Pays de la Loire regions (West of France) mainly in April and August but still persistent from March to August (Figures 2 and 3, box C).

The observed seasonality is related to agricultural practices (fertilizer application period varying
as function of the crop types and farming species) and changes in temperatures, with higher
temperatures favoring volatilization. This explains the high concentration in July and August.
In the Champagne-Ardennes region, areas of hotspots do not correspond to vineyards but to
field vegetables and root crops (from the Institut National de la Recherche Agronomique INRA
https://odr.inra.fr/intranet/carto/cartowiki/index.php/OTEX_et_Orientation_Agricole_des_terri
toires, and AGRESTE, Service Central d'Enquêtes et d'Études Statistiques, 2015
http://agreste.agriculture.gouv.fr/IMG/pdf/R4215A15.pdf). This is a leader region for mineral
fertilization used for sugar industry in France [Ramanantenasoa et al., 2018]. Hamaoui-Laguel et
al. (2014) and Fortems-Cheiney et al. (2016) have previously noted that NH$_3$ emissions in this



region, mainly due to fertilizer over barley, sugar beet, and potato starch in early March, were
higher than what have been reported in the EMEP inventory.
NH$_3$ concentrations are high from April to August in the northern part of the domain that is
known for its animal farming (Eurostat 2014, http://ec.europa.eu/eurostat/statistics-
explained/index.php?title=File:Livestock_density_by_NUTS_2_regions,_EU-28,_2013.png, [Van
Damme et al., 2014a]).
In the Pays de la Loire, NH$_3$ concentrations are high in April and August and remain relatively
high from March to September. Hotspots are found in areas of livestock farming, mainly poultry
and granivorous, which explains the high and relatively constant NH$_3$ concentrations over
warmer periods in this region.
### 3.1.2. Inter-annual variabilities
As can be seen in Figures 2 and 3, NH$_3$ concentrations are enhanced between March and August
in the domain. In this section, inter-annual variabilities are discussed regarding meteorological
conditions and agricultural practices during this time period.
Inter-annual variability of NH$_3$ is higher in springtime than in summer, e.g. in June the variance is
8 times lower than for the other months. To illustrate the inter-annual variability in springtime,
maps of monthly mean NH$_3$ total columns derived in March-April period from IASI (2008-2017
time period) and from CrIS (2013-2017 time period) are shown in Figure 4. Both satellite
distributions exhibit the same inter-annual variability from 2013 to 2017 with higher NH$_3$
concentrations in 2015 over the northern part of the domain than the other years. NH$_3$
concentrations derived from IASI in 2011 are 150% higher in spring (March and April) compared
to 2016 (Figure 4). This inter-annual variability is partly driven by meteorological conditions and
specific agricultural constrains (crop type and phenological stage for instance).
To investigate the impact of meteorological conditions on atmospheric NH$_3$ variability, we
computed the monthly mean anomalies of total precipitation versus skin temperature derived
from ECMWF ERA-interim [Dee et al., 2011], color coded by NH$_3$ total columns anomalies
derived from IASI, as shown in Figure 5. Monthly mean anomalies have been calculated relative
to the 10-years averages (in %). In this figure, monthly NH$_3$ total columns are at least 10% higher
(positive anomalies, red dots) when skin temperatures are higher and total precipitation are
lower than the 10-year average. In contrast, negative monthly NH$_3$ total columns anomalies
(blue dots, Figure 5) are associated with higher total precipitation and lower skin temperatures
than the 10-years average. To further detail this analysis, Figure 1 of the supplement
information shows bar plots of monthly mean NH$_3$ total columns derived from IASI, total
precipitation and skin temperature derived from ECMWF from March to August, plotted in
different colors for the different years of measurements from 2008 to 2017. NH$_3$ total columns





are larger by more than 300% in March-April 2012 compared to 2013 (Figure S1a). Total
precipitation is higher (0.4 mm compared to 1 mm, Figure S1b) and skin temperature is lower
(281 compared to 288 K, Figure S1c) in March 2013 than in March 2012 on average over the
domain. Overall, total precipitation is anti-correlated with $NH_3$ concentrations in the
atmosphere (R = -0.52 from March to August for all years, not shown here) because of a) the
wet deposition importance in the atmospheric $NH_3$ removal and b) the absence of fertilization
during rainy periods. Skin temperature is relatively correlated with $NH_3$ concentrations (R = 0.33
from March to August for all years) since higher temperature increases volatilization of $NH_3$
from the surface to the atmosphere.
In addition, $NH_3$ concentration is maximum in March 2011 whereas it peaks later in April for
2012 (Figure S1a). Springtime is a spreading fertilizer period depending on many agricultural and
meteorological constrains. When temperature are mild, such as in 2012 (Figure S1b), fertilizer
spreading occurs sooner because the phenological growth stage is more advanced. Fertilizing
process period also varies in function of the sowing date which depends on agricultural
practices and crop types: corn is fertilized in early spring whereas rapeseed is in late spring.
Overall, all these meteorological (precipitation and temperature) and agricultural (fertilizer and
manure applications) parameters account for the high $NH_3$ inter-annual variabilities revealed by
both IASI and CrIS in the domain of study.

### 3.2.    Comparisons of $NH_3$ columns derived from IASI, CrIS, and CHIMERE for 2014 and 2015

To discuss the representation of agricultural emissions in the models in terms of intensity and
both spatial and temporal distributions, regional satellite observations derived from IASI and
CrIS have been compared to the CHIMERE model in the region of analysis.

#### 3.2.1. Annual cycle

Standardized monthly mean concentrations derived from IASI, CrIS, and CHIMERE for 2014 and
2015 are shown in Figure 6. These years were selected as $NH_3$ total columns were found to vary
a lot, reaching 10% higher in March and 50% lower in May than the 10-years average
As can be seen from the plot, the 3 datasets exhibit similar patterns in terms of seasonality: all
are enhanced in March-April and in summer, and show a decrease in May. However two major
differences can be noted. First, CrIS standardized $NH_3$ columns are higher in winter (November,
December, and January) compared to the other dataset which can be also be seen in Figure 3.
This could be attributed to a higher number of outliers, given the larger standard deviation
(shaded areas, Figure 6) and no attempt to account for potential non-detects when
concentrations fall below the instrument detection limits. For these months, $NH_3$ levels are low





and undetectable by satellite observations (Figures 2 and 3) so these high values could be
interpreted as observational noise. The detection limit depends on the instrument
characteristics and atmospheric state, with IASI minimum detection limit of ~2-3 ppbv (~4-6.10$^{15}$
molecules.cm$^{-2}$) [Clarisse et al., 2010] and CrIS ~0.5-1.0 ppbv (~1-2.10$^{15}$ molecules.cm$^{-2}$)
[Shephard and Cady-Pereira, 2015; Kharol et. al., 2018]. Second, the CHIMERE standardized NH$_3$
columns are enhanced in September 2014, which is not supported by the observations. It has
been recently shown that CHIMERE overestimated NH$_3$ emissions in autumn over Europe
[Couvidat et al., 2018]. Generally, the amplitude of the modelled seasonal cycle exceeds the
measured ones, which could be explained by higher concentrations measured in winter due to
the observational noise and lower emissions.
Over the whole period, the coefficient of determination (r$^2$) between the standardized monthly
mean NH$_3$ columns derived from IASI (CrIS), and the CHIMERE model is 0.58 (0.18) for the
annual cycles of 2014 and 2015 (not shown here). If we only consider months of high NH$_3$ in the
domain from March to August, the correlation between the observational datasets and the
model is good with linear regression slope values between IASI (CrIS) and CHIMERE of 0.98
(0.71), as shown in Figure 7. The seasonal cycle is thus well reproduced by the model, which is
encouraging given the fact that annual total emissions are simply disaggregated with a monthly
profile in the model. However, the values of the r$^2$ lower than 0.5 indicate that the CHIMERE
model only reproduces at most half of the observed monthly temporal NH$_3$ variabilities in the
domain. Similar variabilities are found between the observations and the model outputs since
the coefficients of correlation of the standard deviations are 0.4 and 0.6 between CHIMERE and
IASI and CrIS, respectively.
### 3.2.2. Spatial variability of NH$_3$ in springtime
The IASI and CrIS regional maps have been compared to the CHIMERE model for the March-April
period in 2014 and 2015 to evaluate the model's capacity to reproduce the spatial distribution
of the episodic emissions from fertilizer spreading practices in springtime, as well as their inter-
annual variability. Satellite NH$_3$ measurements in springtime have been gridded at 0.15°x 0.15°
spatial resolution, and the associated CHIMERE maps have been computed following the
coincident criteria described in section 2.4 at the same spatial resolution (Figures 8 and 9).
First one can notice that the spatial distribution of NH$_3$ observed in springtime by both satellite
instruments are in good agreement, even though their overpass time is different (~4 hours
apart). This was already seen in the inter-annual variability agreement seen in Figure 4. In spring
2014, IASI and CrIS both reveal three main regions of enhanced NH$_3$ concentrations (North,
Champagne-Ardennes, and Brittany/Pays de la Loire region) already identified by the 10-years
and 5-years of IASI and CrIS observation maps (Boxes A, B, and C of Figures 2 and 3). In 2015,
concentrations of NH$_3$ in the northern part of the domain are higher than in 2014, as indicated



by both IASI and CrIS observations (Figure 9, upper panels). Overall, satellite observations are
able to capture similar spatial distributions of high $NH_3$ concentrations in springtime, and their
evolution in time.
In spring 2014, the CHIMERE model reproduces the high concentrations in the three regions of
the domain identified in Figures 2 and 3. Additional $NH_3$ hot spots in the southeastern part of
the domain including the Po Valley, Switzerland, and the wine region between Besancon and
Lyon (blue box in Figure 8) are indicated by the CHIMERE model. $NH_3$ emissions in this latter
region are comparable to average agricultural plains over France. Only dispersion conditions
related to wind speed and boundary layer height can explain high $NH_3$ concentrations over this
area.
In spring 2015, satellite observations and the CHIMERE model outputs exhibit very similar
patterns in term of high $NH_3$ distributions, with however higher $NH_3$ concentrations indicated by
the model in the southern part of the domain (blue box in Figure 9).
Finally, the (model - observations) differences between the standardized $NH_3$ column derived
from the satellite instruments in springtime 2014-2015 and the corresponding $NH_3$ columns
derived from the CHIMERE model are shown in Figure 2 of the supplement information. One
can see that very similar patterns are presented when comparing the model to independent
satellite observations from IASI and CrIS: the modelled $NH_3$ concentrations are systematically
lower for both years over Belgium and United Kingdom, and higher in the southern part of the
domain (green square, Figure S2) including the Pays de la Loire region (box C in Figures 2 and 3),
and in the southeastern part of the domain (over the North part of Switzerland and the Plateau
du Jura region - between Besancon and Lyon cities – blue box in Figure 8). Reasons of enhanced
$NH_3$ columns derived from the model in this latter region are not clear yet. An explanation could
be that the temporal distribution of the emissions is misrepresented in the model since the
modelled concentrations are enhanced in April whereas the two satellite observations are
enhanced earlier in March for both years. It is worth noting that there are no EMEP stations
measuring surface $NH_3$ concentrations in these regions. As for the Brittany/Pays de la Loire
region, it has already been shown that the LOTOS-EUROS atmospheric model [Schaap et al.,
2008] using similar chemistry schemes and $NH_3$ emissions shows higher columns each year in
this area [Van Damme et al., 2014b].

### 3.3. Conditions for PM formation in the Paris megacity

To investigate the impact of intensive agriculture practices on the Paris megacity air quality, we
need to better understand the role of $NH_3$ in the formation of $PM_{2.5}$ that depends, among
others, on specific meteorological conditions such as atmospheric temperature and humidity
that alter the gas-particle partitioning. The link between high $NH_3$ concentrations inducing $PM_{2.5}$


formation in the Paris megacity is known [Petetin et al., 2016; Zhang et al., 2013] but
quantification of such phenomena is difficult due the lack of long-term $NH_3$ monitoring in the
IdF region. $PM_{2.5}$ is however measured hourly at several locations in Paris by the Airparif
network (https://www.airparif.asso.fr/, Figure 1). Thanks to the 10 years of IASI observations,
an observational evidence of $PM_{2.5}$ formation in the IdF region (100 km around Paris - black box
in Figure 1) is represented in Figure S3. Simultaneous enhancements in March of $PM_{2.5}$
measured at the surface and $NH_3$ columns derived from the IASI observations over the IdF
region are clearly visible. However, high concentrations of $NH_3$ observed in summer are not
associated with high $PM_{2.5}$ concentrations. This reflects the complexity of the $PM_{2.5}$ formation
depending on various factors, such as $NH_3$ emissions, atmospheric chemistry (acidic content of
the atmosphere), transport, and specific meteorological conditions involved in the gas to solid
phase conversion between $NH_3$ and ammonium salts.
To evaluate the impact of long-range transport on $NH_3$ levels observed over the Parisian region
(IdF) in spring, back-trajectory analysis was performed. In total 231 24-hours back-trajectories
ending in Paris (period from February 15[th] to May 15[th] from 2013 to 2016) were classified into 8
clusters using HYSPLIT (https://ready.arl.noaa.gov/HYSPLIT.php). Figure 10 shows the mean
trajectories for each cluster associated with the average $NH_3$ total columns measured by IASI
over the IdF region. In this figure, higher $NH_3$ columns are found under the influence of air
masses transported from the northern part of the domain (over Belgium and the Netherlands,
clusters 4 and 5) and from the Brittany region (cluster 8), which are the major sources regions of
$NH_3$ in spring in the domain as previously identified (Figures 2 and 3). Clusters 2 and 3 (Figure
10) are associated with intermediate $NH_3$ levels since air masses moved slowly transporting
$NH_3$-rich air from rural regions near IdF (such as the Champagne-Ardennes region - Box A in
Figures 2 and 3) to Paris. Finally, low $NH_3$ concentrations are measured when air masses
originated from ocean regions passing through continental areas with minor $NH_3$ sources in
spring (clusters 1, 6 and 7, Figure 10). This reflects the importance of long-range transport in the
$NH_3$ budget observed over the Paris megacity in spring.
To quantitatively assess the influence of meteorological parameters on the formation of $PM_{2.5}$
from $NH_3$ in the IdF region, timeseries of $NH_3$ total columns, $PM_{2.5}$ surface concentrations, and
four meteorological parameters (temperature at 2 m, boundary layer height, total precipitation
and relative humidity) derived from ECMWF - ERA-Interim [Dee et al., 2011] were analyzed. To
compute daily and monthly means, IASI $NH_3$ total columns have been averaged over IdF (black
box in Figure 1), $PM_{2.5}$ concentrations measured between 9 AM and 11 AM have been averaged
over the 14 stations (dark points in Figure 1), and ECMWF data have been averaged over a 300
km region around Paris (the blue box in Figure 1). Figure 11 shows all these parameters for
spring 2014.



We have flagged pollution episodes in both time series ($PM_{2.5}$ and $NH_3$) by selecting data above
1-sigma standard deviation over the mean of the datasets from 2013 to 2016. This time period
was selected to have enough IASI observations in the IdF region. Then two cases have been
defined to study the temporal correlation between $NH_3$ and $PM_{2.5}$: case A in which both $NH_3$ and
$PM_{2.5}$ pollution episodes appear simultaneously, i.e. within the same day or 2 days apart
(shaded in red in Figure 11); case B in which pollution episodes appear at least 3 days apart
(shaded in blue in Figure 11). In Figure 11, a strong relationship between peaks of $NH_3$, $PM_{2.5}$
and meteorological parameters can be seen. For example, between March 3$^{rd}$ and March 19$^{th}$
2014 (case A), the boundary layer height is exceptionally low (456 m; compared to 760 m on
average); the temperature is relatively low (280 K; 282 K on average); and there is no
precipitation (0.01 mm/h; 0.11 mm/h on average). One note that peaks of maximum $NH_3$
observed in IdF on March 11$^{th}$ and 12$^{th}$ are associated with air masses coming from the northern
part of the domain (clusters 4 and 5 in Figure 10). In contrast, for the case B in which
appearance of peaks of $NH_3$ and $PM_{2.5}$ is not simultaneous, meteorological conditions are
different: the boundary layer is thicker (908 m on April 23$^{rd}$ 2014), or temperature is higher (285
K on April 11$^{th}$ 2014).
To further investigate the influence of meteorological parameters on the pollution episodes in
the IdF region, detailed analysis have been made over the whole dataset. Figure 12 shows the
statistical distribution of meteorological parameters corresponding to case A, case B, and all
observations. One can see that the boundary layer height is significantly lower in case A (550 ±
205 m) than in case B (751 ± 276 m), and that precipitations are absent in case A (0,019 mm/h)
compared to case B (0,085 mm/h). The temperature at 2 meters also differs between the two
cases (case A: 278 ± 3 K; case B: 282 ± 4 K), but the humidity is almost the same (70% ± 17%
versus 75%±18%). Thus the combination of the following three meteorological parameters favors
simultaneous appearances of $NH_3$ and of $PM_{2.5}$ in Paris (i.e. case A): low surface temperatures
(5°C), with thin boundary layers (~500m), and rare precipitations. In addition, the Wilcoxon-
Mann-Whitney test ([Wilks, 2011], not shown here) indicates that each single parameter has no
significant influence on the $NH_3$-$PM_{2.5}$ correlation. Therefore only a combination of these
different parameters has an impact on secondary aerosol formation from $NH_3$.
An explanation of these findings might be that anticyclonic conditions (low planetary boundary
layer), preventing pollutant dispersions in the lower atmosphere [Salmond and McKendry,
2005], along with moderate wind fields allow $NH_3$ plumes to be transported from rural to urban
regions [Petit et al., 2015]. In addition, thanks to relatively low atmospheric temperatures and a
moderate relative humidity, conversion of gas phase $NH_3$ to ammonium salts is then
accentuated via optimal phase equilibrium [Watson et al., 1994; Nenes et al., 1998]. Finally,
with the absence of rain, ammonium salts are stabilized in the aerosols.


Our observations are in agreement with previous studies [Bessagnet et al., 2016; Wang et al.,
2015], which have shown that the formation of ammonium salt needs a specific humidity of 60 -
70%, because it corresponds to the deliquescence point of $NH_4NO_3$ in ambient air. This is in
agreement with our results since the mean of relative humidity in case A is 70%. Our results also
support the idea that a relatively low atmospheric temperature favor $PM_{2.5}$ formation since the
phase equilibrium leads to $NH_4NO_3$ decomposition above 30 °C.

## 510  4. Conclusions

This study focuses on seasonal and inter-annual variabilities of $NH_3$ concentrations in a 400 km
radius-circle area around Paris to assess the evolution of major $NH_3$ agricultural sources and its
key role in the formation of the secondary aerosols that affect air quality over the Paris
megacity.
Thanks to 10-years and 5-years of regional $NH_3$ observations derived from IASI and CrIS, three
main regions of high $NH_3$ occurring between March and August were identified. Observed inter-
annual variabilities of $NH_3$ concentrations have been discussed with respect to total
precipitations and atmospheric temperature, showing that total precipitations are anti-
correlated with high $NH_3$ concentrations, and that mild temperature in late winter causes
precocious fertilizer spreading due to advanced phenological growth stage.
To evaluate our knowledge on agricultural emissions in terms of intensity and both spatial and
temporal distributions, coincident CHIMERE model outputs have been compared to satellite
observations of IASI and CrIS for 2014 and 2015. The annual cycle is well reproduced by the
model (correlation slopes of 0.98 and 0.71 between the model and IASI and CrIS, respectively)
but the model is only able to reproduce half of the observed atmospheric $NH_3$ variability.
Focusing on spring periods (March-April 2014 and 2015) of episodic $NH_3$ emissions, the two
independent satellite observations derived from IASI and CrIS show very similar spatial
distributions of high $NH_3$ concentrations, as well as their evolution in time. The comparison
between CHIMERE $NH_3$ columns and coincident satellite observations highlights the same
difference spatial patterns with a systematic underestimation of $NH_3$ concentrations from the
model over Belgium and an overestimation in the southern part of the domain (French
Brittany/Pays de la Loire and Plateau du Jura regions, as well as North of Switzerland).
Focusing on the Ile-de-France (IdF, 100 km around Paris) region, we found that air masses
originated from rich-$NH_3$ areas, mainly the northern part of the domain over Belgium and the
Netherlands, increase the observed $NH_3$ total columns measured by IASI over the urban area of
Paris.
To assess the link between $NH_3$ and $PM_{2.5}$ over the Parisian (IdF) region, the main
meteorological parameters driving the optimal conditions involved in the $PM_{2.5}$ formation have





been identified. The results show that relatively low temperature, thin boundary layer, coupled
with almost no precipitation, favor the PM$_{2.5}$ formation with the presence of atmospheric NH$_3$ in
the IdF region. Based on a more observational approach over large time scale, this work is in
agreement with previous studies.
This study highlights the need for a better representative NH$_3$ monitoring to improve numerical
simulation of spatial and temporal NH$_3$ variabilities, especially at fine scales. In order to
compare IASI and CrIS data in absolute values, it would be recommended to derive both
datasets using the same retrieval algorithm. Thus, by combining these datasets bi-daily NH$_3$
total columns in absolute values at regional scale would be provided. This would help inferring
variability of top-down NH$_3$ emissions. Complementarily, long term quantification of NH$_3$ diurnal
cycle inside Paris would improve comparisons with local PM$_{2.5}$ needed to understand secondary
aerosols formations. For this purpose, an ongoing activity consists in the deployment of a mini-
DOAS instrument [Volten et al., 2012] used for long-term and continuous monitoring of
atmospheric NH$_3$ concentrations in the center of Paris from the QUALAIR platform
(https://www.ipsl.fr/en/Our-research/Atmospheric-chemistry-and-air-quality/Tropospheric-
chemistry/QUALAIR). Finally, the geostationary-orbit sounder IRS-MTG ([Stuhlmann et al.,
2005], to be launched after 2022) will provide NH$_3$ columns at very high sampling rate (every 0.5
hour over Europe) with an unpreceded spatial resolution (pixel size of 4 km).

Author contribution:
CV wrote the paper with contributions of all coauthors. CV and CC designed the study. MV, LC,
and SW performed IASI retrievals and ED, MWS, and KEC performed the CrIS retrievals. FM ran
the CHIMERE simulations. CV and TW analyzed the data with guidance from CC and PFC. All
authors discussed the results and contributed to the final paper.
Acknowledgement:
IASI is a joint mission of Eumetsat and the Centre National d'Etudes Spatiales (CNES, France).
This work was supported by the CNES. It is based on observations with IASI embarked on
Metop. The IASI Level-1C data are distributed in near real time by Eumetsat through the
EumetCast system distribution. The authors acknowledge the Aeris data infrastructure
(http://iasi.aeris-data.fr/NH3/) for providing access to the IASI Level-1C data and Level-2 NH$_3$
data used in this study. The French scientists are grateful to CNES (TOSCA) and Centre National
de la Recherche Scientifique (CNRS) for financial support. The research in Belgium is also funded
by the Belgian State Federal Office for Scientific, Technical and Cultural Affairs and the European
Space Agency (ESA Prodex IASI Flow project). The CrIS Fast Physical Retrieval (CFPR) NH$_3$ data is
provide through a joint collaboration between Environment and Climate Change Canada (ECCC)
and Atmospheric and Environmental Research (AER), Inc. (USA). The Level 1 and Level 2 input
data for CFPR were obtained from the University of Wisconsin-Madison Space Science and



Engineering Center (SSEC) and the NOAA Comprehensive Large Array-Data Stewardship System
(CLASS) (Liu et al.,2014), with special thanks to Axel Graumann (NOAA).

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



# FIGURES


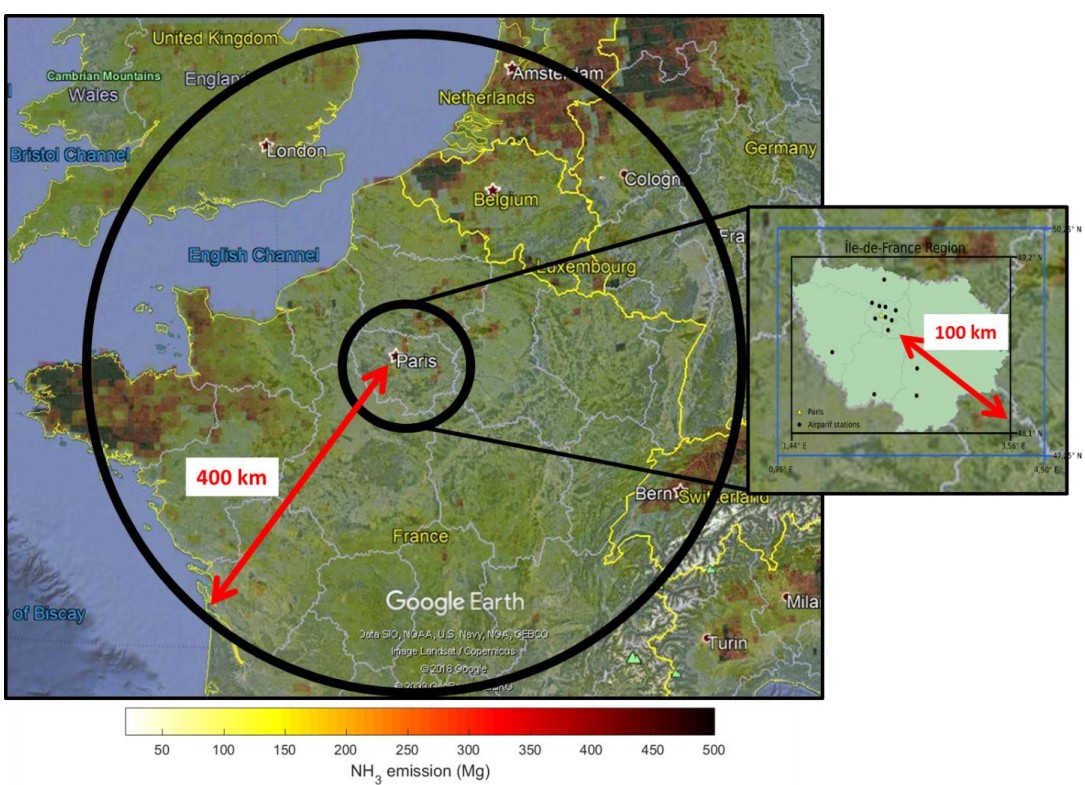


Figure 1: Region of analysis: 400 km radius-circle around the Paris megacity and 100 km around
Paris. The latter is representative of the Ile-de-France (IdF) region where the Airparif PM
observational network is located. Black points are the locations of the Airparif stations
measuring hourly $PM_{2.5}$ concentration at the surface. The black (blue) box delimitates the IdF
region in which the IASI $NH_3$ (ECMWF) data have been considered. The overlay represents $NH_3$
emissions (in Mg per year and per cell of 0.1°x0.1°) derived from the EMEP inventory for 2015.





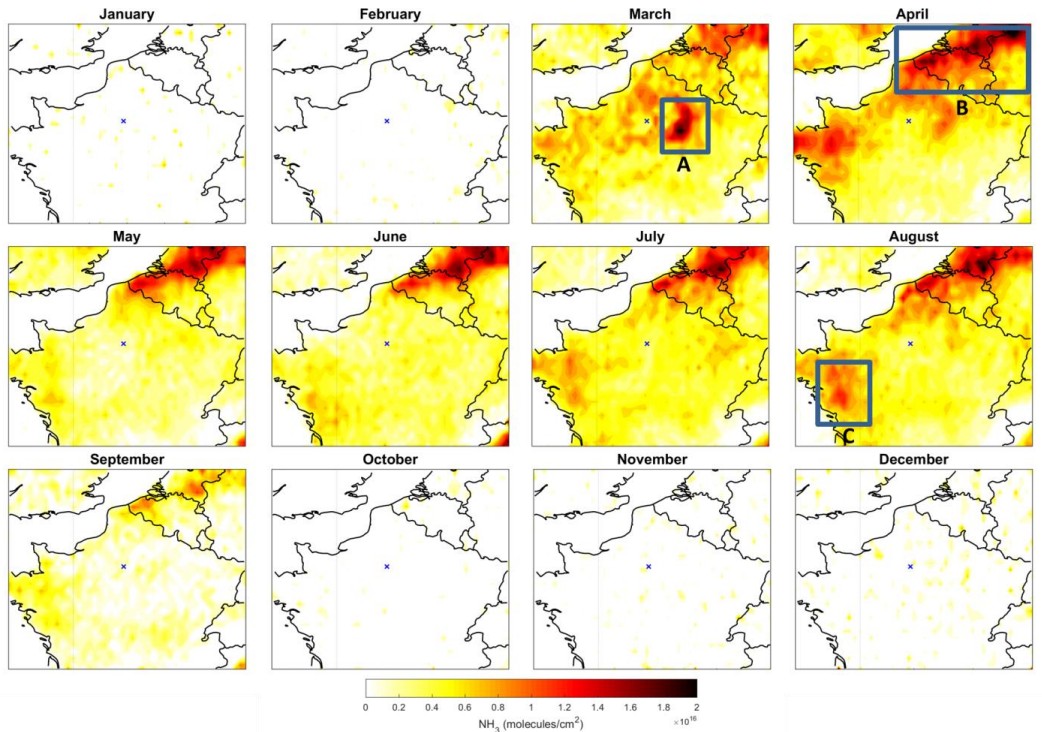


Figure 2: Monthly means of NH$_3$ total columns (molecules/cm²) derived from 10 years (2008-2017) of IASI NH$_3$-retrieved columns. The blue cross indicates Paris location.






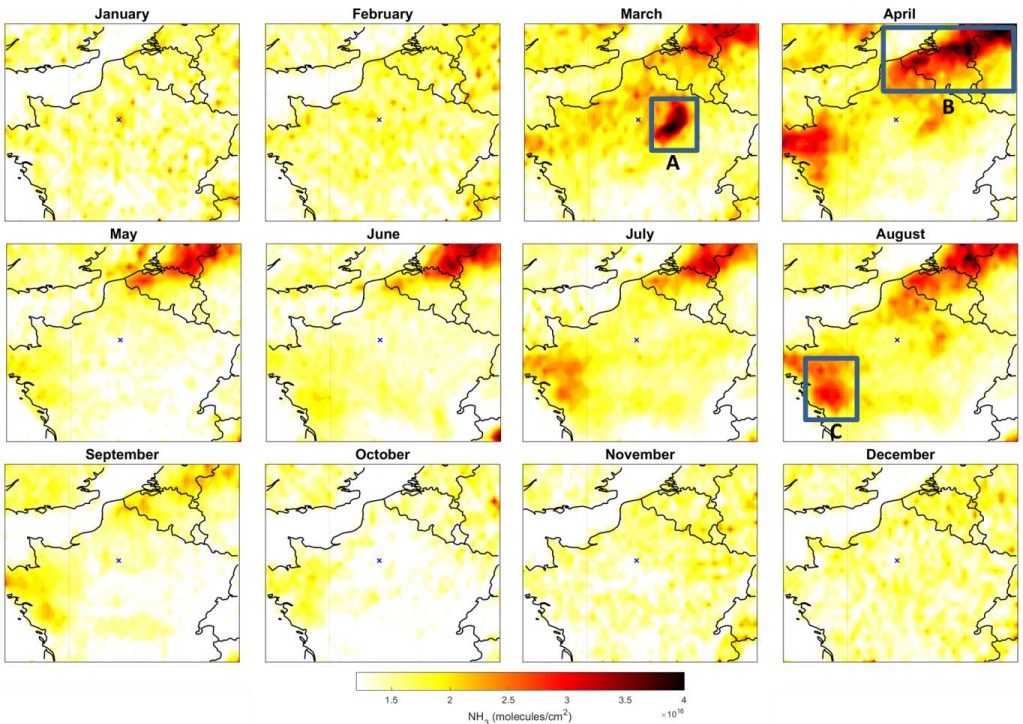


Figure 3: Monthly means of NH$_3$ total columns (molecules/cm²) derived from 5 years (2013-2017) of CrIS NH$_3$-retrieved columns. The blue cross indicates Paris location.



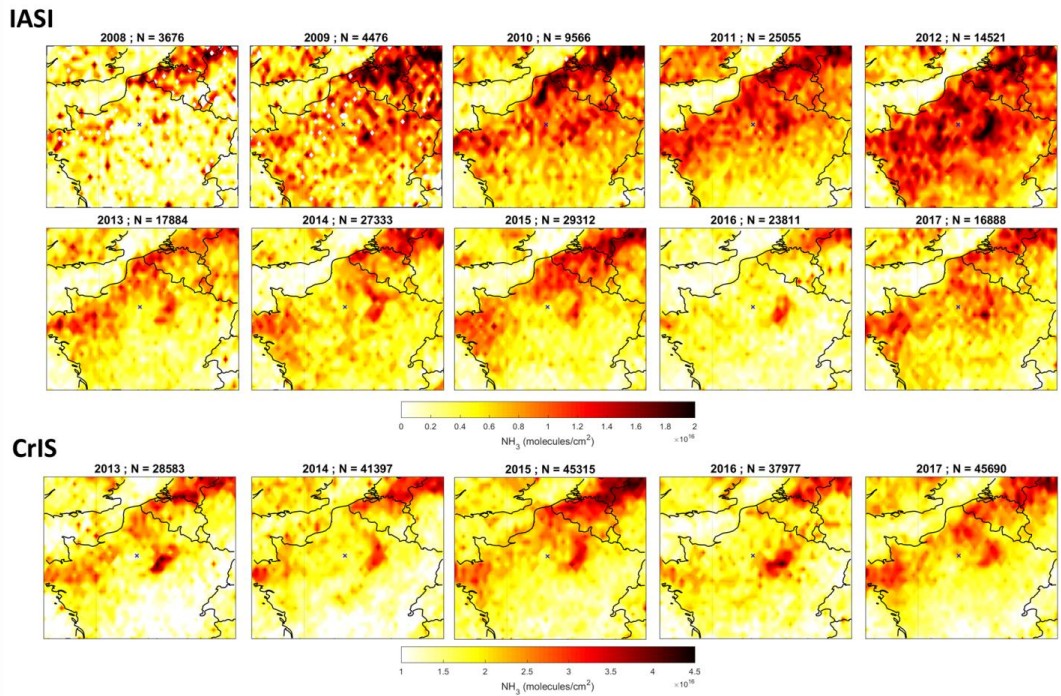


Figure 4: Maps of monthly mean NH$_3$ total columns (molecules/cm²) in March-April period derived from IASI from 2008 to 2017 and CrIS from 2013 to 2017.



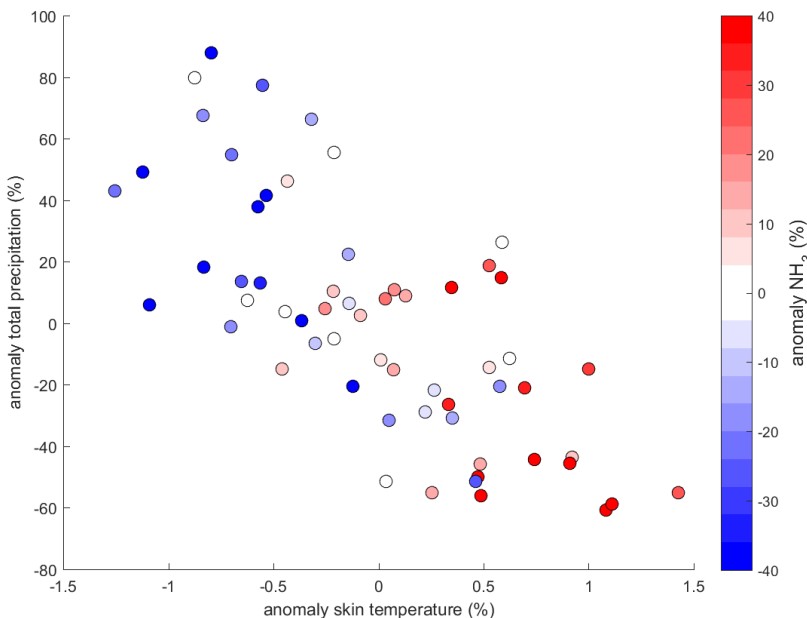


Figure 5: Scatter plot of monthly mean anomaly (relative to the 10-years – 2008 to 2017 - monthly average) of total precipitation versus skin temperature derived from ECMWF from March to August in the domain, and color coded by the NH$_3$ total columns anomaly derived from IASI.





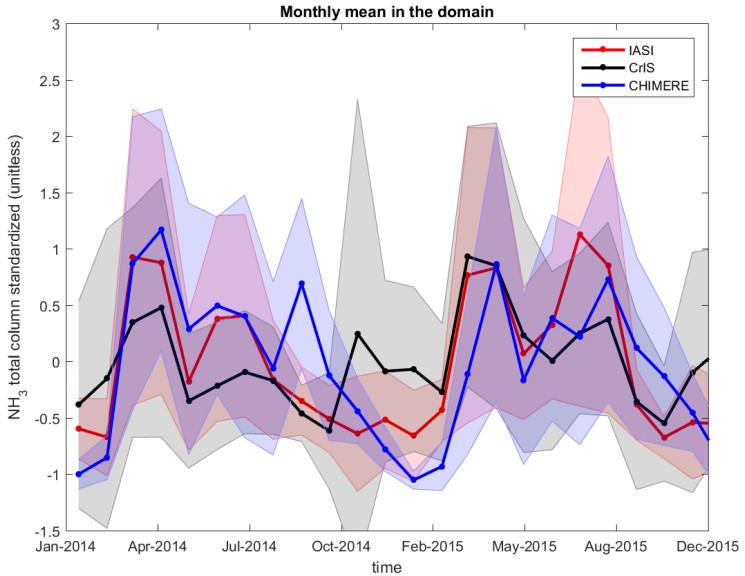


Figure 6: Standardized monthly mean concentrations derived from IASI (red), CrIS (black) and
CHIMERE (blue) for 2014 and 2015. Shaded areas correspond to the one-sigma standard
deviation around the means.





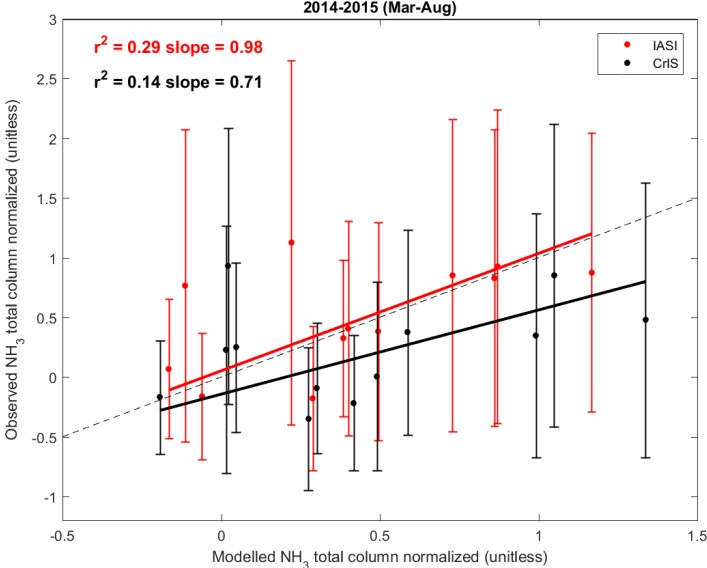


Figure 7: Correlation plots between monthly means $NH_3$ standardized concentrations derived from satellite observations (IASI in red and CrIS in black) and the CHIMERE outputs for the March to August months of 2014 and 2015. The 1:1 line is represented in the dashed line.



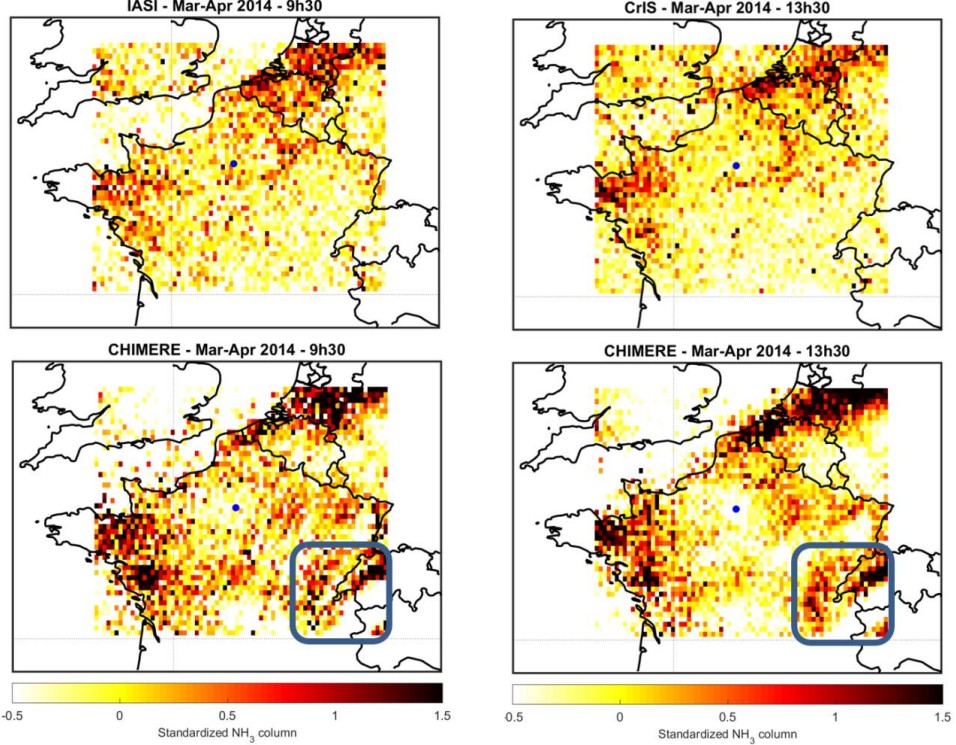


Figure 8: Standardized NH$_3$ column derived from the satellite instruments (IASI = top left panel, and CrIS = top right panel) and the corresponding NH$_3$ column derived from the CHIMERE model (coincident with IASI – bottom left panel, and coincident with CrIS – bottom left panel) for March-April 2014. Blue dots indicate Paris location.



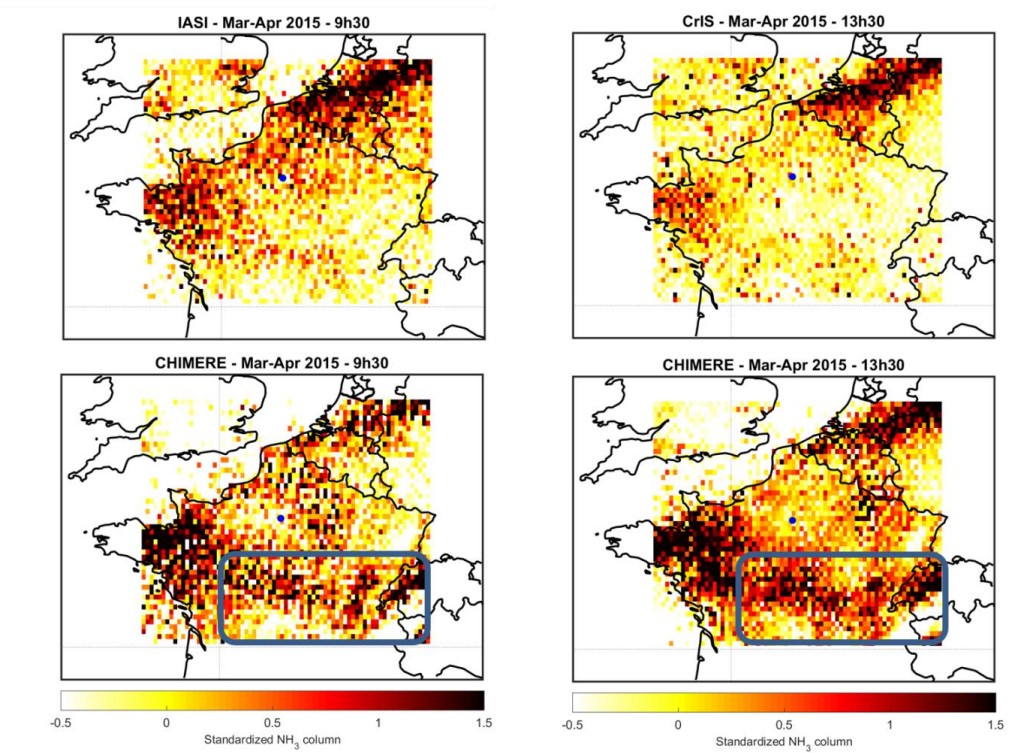


Figure 9: Same than Figure 7 but for March-April 2015.




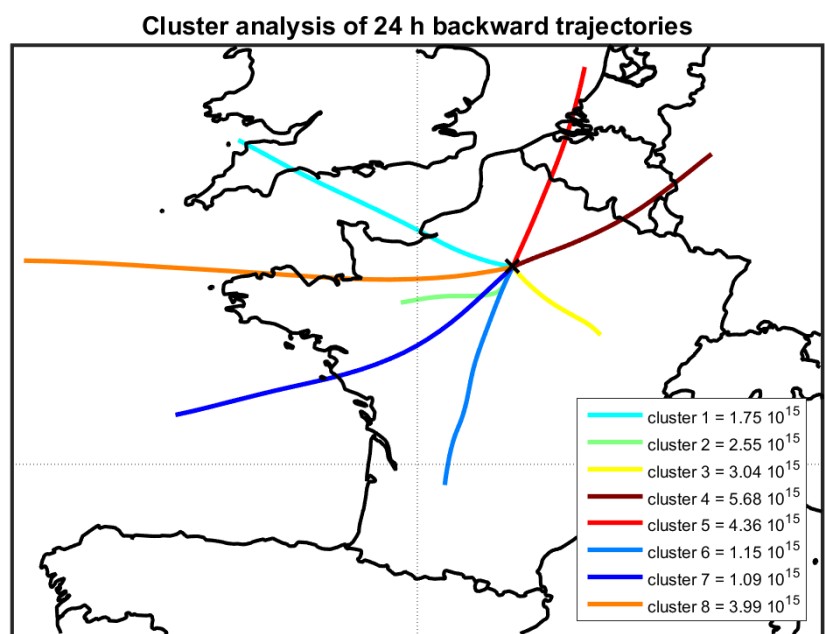


Figure 10: Cluster analysis of 24-h backward trajectories arriving in spring in Paris (from
February 15th to May 15th for the 2013-2016 period) using HYSPLIT-4 model obtained from the
NOAA Air Resources Laboratory. Mean trajectories of the 8 clusters are shown in different
colors, associated with the $NH_3$ concentrations measured by IASI in the IdF region (in
molecules/cm$^{-2}$).





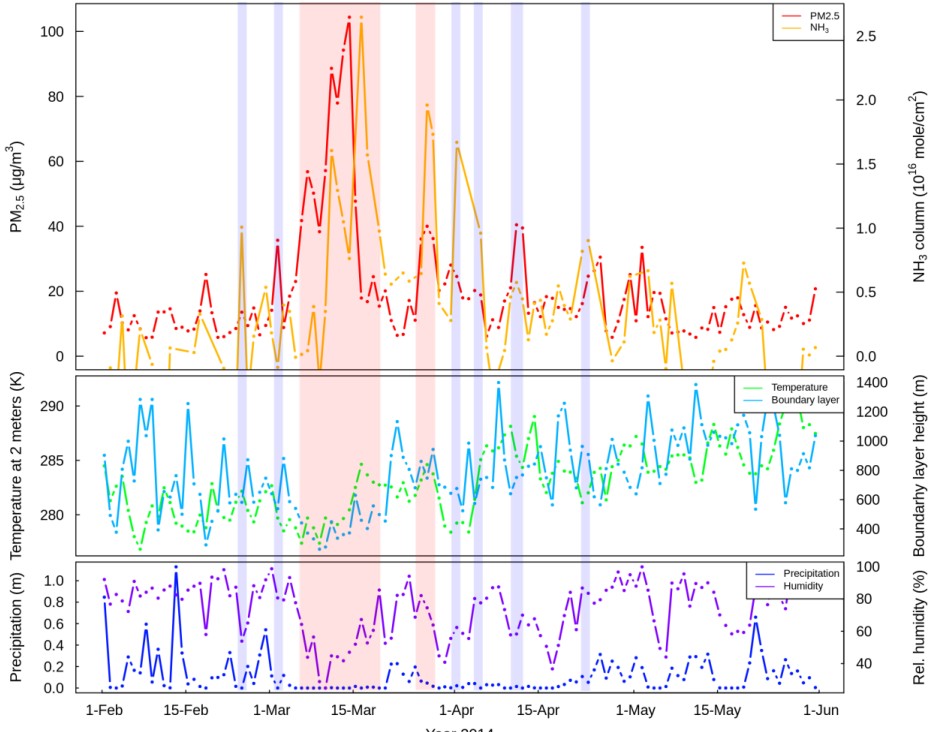

Figure 11: Average concentrations of $NH_3$ total columns derived from IASI (in molecules/cm²;
orange, upper panel) and $PM_{2.5}$ derived from the Airparif network selected within 2 hours from
the IASI overpass (in µg/m³; red, upper panel) for 2014 as example. Periods of simultaneous
(independent) enhancements of $NH_3$ and PM concentrations are represented with red (blue)
areas, i.e. case A (case B). Temperature at 2 meters (in Kelvin; green, middle panel), boundary
layer height (in meter; blue, middle panel), precipitation (in meter; dark blue, lower panel), and
relative humidity (in percent; purple, lower panel) derived from the ECMWF ERA-interim.





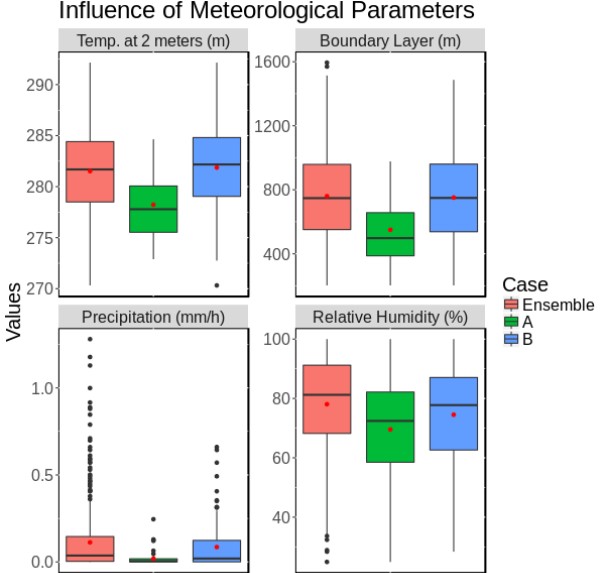

Figure 12: Statistical distributions of meteorological parameters corresponding to case A, case B,
and all observations derived from 2013 to 2016. The medians and the quartiles are presented by
center lines and borders of the boxes, respectively. The mean values are indicated by red points,
and the extreme values (i.e. those beyond Q1 - 1.5 IQR and Q3 + 1.5 IQR) by black points.



**TABLE**

| | Satellite | Overpass time (LT) | Time coverage | Nadir spatial resolution (km) | Spectral range (cm⁻¹) | Spectral resolution (cm⁻¹) | Spectral Noise[*] (K) @270K @ 970 cm⁻¹ | References |
|---|---|---|---|---|---|---|---|---|
| **IASI** | Metop-A/B | 9.30 (AM/PM) | 2006-present | 12 | 645–2760 | 0.5 (apodized) | ~0.2 | Clerbaux et al., 2009 |
| **CrIS** | Suomi-NPP | 1.30 (AM/PM) | 2011-present | 14 | 645–1095; 1210–1750; 2155–2550 | 0.625; (unapodized) | ~0.05 | Zavyalov et al., 2013 |

[*]Spectral noise comparison values in main ammonia spectral region (~970 cm⁻¹) obtained from Zavyalov et al., 2013.

Table 1: Instrumental specifications for the IASI and CrIS satellite instruments.