# Peer review of "Atmospheric ammonia variability and link with PM formation: a"

_Atmospheric Chemistry and Physics, 2019_

## Referee Comment (RC1) · Anonymous Referee #2 · 29 Apr 2019

General comments

In this study, Viatte et al. use satellite observations (CrIS, IASI) to a) characterize the spatial and inter annual variability of ammonia column over Western Europe and its drivers and b) examine the connection between NH3 and PM2.5 over Paris. The material presented is interesting and well suited for ACP However, I have some significant concerns regarding the robustness of some of the conclusions and the lack of connection between a) and b). These need to be addressed before I publication can be considered.

General Comments

a) there are places when the authors make fairly definitive claims with insufficient sup-

port/references. For instance Line 49: it is stated that N causes species/ecosystem extinction. A specific reference is needed. Line 341 and discussion above. This discussion is too speculative and needs to be much better supported. Was more corn planted in 2011 than in 2012? Were planting dates shifted earlier in 2011 relative to 2012? This is critical since the authors then state that they have shown that meteorology and farming practices account for the interannual variability in NH3 column. Line 374 It is stated that the correlation is "good" based on Fig. 7 (r2<0.3) . What is the p value, what is the uncertainty on the slopes given the large error bars shows in Fig. 7? In general, the authors need to be more quantitative when reporting statistics: always give p value for correlation (e.g ., line 331 and 333) and uncertainty for slopes.

b) there is very little connection between a) and b) in the current manuscript. In part b), the authors focus on the relationship between PM2.5 and NH3 in two (fairly similar) years (2014,2015). The main conclusion is that meteorology (temperature, local BL) probably controls whether NH3 contributes to PM2.5. This is interesting although very much expected from studies performed in other regions. From part a), I was instead expecting the authors to consider whether the considerable variability in NH3 sources over Belgium/Netherlands could impact PM2.5 over Paris. From part a), I was also expecting to have the authors show whether CHIMERE is able to capture the observed correlation between PM2.5 and NH3. This could help understand whether the observed PM2.5 enhancement results from production of ammonium nitrate in Ile de France or from transport of ammonium nitrate/sulfate or other aerosols from Belgium. I fully appreciate that such analysis will require significant work. However, without a significantly stronger connection between part a) and b), I would recommend the paper be split, with part a) being more readily publishable.

Technical comments

Section 2.3 the description of CHIMERE is far too short (especially with respect to the treatment of ammonia. For instance: -> how is dry deposition represented? Does it include the bidirectional exchange between land and atmosphere -> what is the temporal

resolution of the emissions? Does it include a diurnal cycle? It would be useful to show the seasonality of the emissions in a few regions, to help the reader better analyze Figs 2 and 3 -> how is the gas/aerosol partitioning of NH3 represented (ISORROPIA?) -> I assume that NH3/NH4/NH4NO3 in CHIMERE have been evaluated previously? Please provide reference for these studies at this stage. I also encourage the authors to show how the configuration of CHIMERE that is used here performs against surface observations (e.g., EMEP wet deposition/concentrations). This could be briefly discussed in the main text, with figures in the supplementary materials.

Section 3.1.1

It would be useful to include a map showing the distribution of livestock and major crops in Western Europe so that the reader can see the relationship between NH3 emissions and the different sources described by the authors.This would be especially helpful as some of the material the authors refer to is in French.

Fig. 5. This figures shows first and foremost that there is good correlation between skin temperature and precipitation at the regional level. I think it would be more relevant to show the relationship between temperature/precipitation and NH3 anomaly In addition, I assume that the precipitation/temperature anomalies exhibit some significant spatial variability? Do you weigh the anomaly by the average NH3 column? High NH3 columns only cover a small fraction of your domain and it's unclear to me why it would respond to the average temperature change (vs the local change)

Section 3.2

I am a little confused by the need for the standardization. CrIS and IASI seem reasonably close, so why not use the model absolute NH3 column. In addition, Fig. 6 only show one CHIMERE time series, shouldn't there be two, one for CHIMERE sampled at the IASI overpass time and one at the CrIS overpass time (with AK)..

Line 351 I am not sure I understand the motivation for picking this years. Why not use

the climatological seasonality? Why are these years more useful to benchmark the model? They look fairly similar as far as I can tell from the supporting material.

Technical comments

They are a few issues with language. It sometimes (rarely) makes it challenging to understand the manuscript line 28: regression slope. Remove slope line 63: related->relative Line 112: many of studies? Line 283: farming species? Do you mean livestock? Line 300. What are non-poultry granivorous (animals)?

Fig. 7 What do the error bars correspond to?

Fig. 9: Same than Fig.7 -> "Same as Fig. 8"

Fig. 12: Define IQR

Line 220: I don't understand the distinction between inorganic, organic and natural aerosols?

Line 487. Why is the value given on line 476 different (mean/median?)

---

## Referee Comment (RC2) · Anonymous Referee #1 · 5 Jul 2019

This manuscript uses observations from two independent satellites to assess the role of NH3 in springtime particle pollution episodes in the Paris region by examining the seasonal and interannual variability (IAV) in NH3 columns over northwestern Europe. The observations are compared to simulations from the CHIMERE chemical transport model. In general, the authors do a good job of reviewing the existing literature to provide context for their results, but it would be useful if they could include a comparison with the study of Schiferl et al. (2016), which examines seasonal cycles and IAV of NH3 over the US.

In Section 2.2, it is important that the authors report what proportion of the column observations from each satellite were below the limit of detection and how those data were incorporated into the monthly means used throughout the paper. If observations

below the limit of detection were discarded, then the resulting monthly means will be biased high. It would then be important to filter the model output in a similar way to ensure that the observation-model comparison is more appropriate.

A general concern in Section 3.1 is the confidence with which the authors interpret the causes contributing to the seasonality and IAV of the ammonia columns. In many cases, the explanations provided by the authors seem reasonable, but unless there is conclusive proof, the language should be toned down to indicate that these are possible/likely explanations rather than the only ones:

Lines 282-301, a handful of data are provided to describe farming practices in different regions, but not in a consistent way. What evidence is there that the factors described are the most important in causing the spatial and temporal patterns observed?

Lines 313-314 How do crop type and phenological stage impact ammonia concentrations leading to interannual variability?

Lines 330-333 These seem like plausible explanation for the impact of precipitation amount of ammonia columns, but is there direct evidence that they are the only (most) important factors?

Lines 334-335 The relationship between gas phase ammonia and temperature should be exponential based on the temperature dependence of its volatilization (either vapor pressure or effective solubility). Does the correlation coefficient change if a non-linear fit is tried?

In Section 3.2, the authors compare 'standardized' monthly means for the years 2014 and 2015 between the two satellite products and the model. More explanation should be provided about how these standardized means were calculated. Do the emissions used in the model differ between the two years? This would be useful to know to help in interpreting the variability produced by the model.

Lines 371-382 This discussion is a bit confusing because initially the values quoted

from the correlation plots of are the coefficients of determination, and then the comparison is restricted to select months and the values quoted are the slopes. I would recommend quoting the r2 values for both, to make it more clear that the coefficients of determination did not increase significantly when the months were restricted. Also, the fact that the slope is close to 1 is not that meaningful since each dataset has already been standardized.

In Section 3.3, which focuses on the role of NH3 in producing PM2.5 in the Ile de France region, the analysis is overly simplistic. Why have the PM2.5 observations included in the analysis been restricted the measurements between 9 and 11 am? This time interval is particularly challenging to interpret because of the impacts of primary emissions and the role of the rapidly changing boundary layer height. It seems like a poor choice of time window to focus on a phenomenon that is influenced by long-range/regional transport of a precursor species like NH3. The role of temperature and relative humidity on the formation of ammonium salts is well-described by thermodynamic relationships. Statements like those on Lines 504-509 and not fully accurate.

Specific comments: Line 46 – 'biochemical' should perhaps be 'biogeochemical'

Line 63 – 'related to' should be 'relative to'

Line 111-114 – It would be helpful to reword the sentence slightly, to clarify that all of the studies being referenced were carried out in Paris.

Figure 1 – The coloring of the map by the emissions is not easy to see. The colors become a very different shade on the map then on the legend. Is it possible to use a map that doesn't have a green background, or to make the emissions coloring more opaque?

Figure 6 – would be helpful to have the same months identified on the axis for each year

References
Luke D. Schiferl, Colette L. Heald, Martin van Damme, Lieven Clarisse, Cathy Clerbaux, et al. Interannual variability of ammonia concentrations over the United States: sources and implications. Atmospheric Chemistry and Physics, 16 (18), 12305-12328, 2016.
* * *

---

## Author Comment (AC1) · 4 Sep 2019

"Atmospheric ammonia variability and link with PM formation: a case study over the Paris area" by Camille Viatte et al.

Anonymous Referee #1 Authors: We would like to thank the referee for his/her insightful comments. We have made changes to the manuscript to address those comments.

Referee: This manuscript uses observations from two independent satellites to assess the role of NH3 in springtime particle pollution episodes in the Paris region by examining the seasonal and interannual variability (IAV) in NH3 columns over northwestern Europe. The observations are compared to simulations from the CHIMERE chemical transport model. In general, the authors do a good job of reviewing the existing lit-

erature to provide context for their results, but it would be useful if they could include a comparison with the study of Schiferl et al. (2016), which examines seasonal cycles and IAV of NH3 over the US. Authors: We have added sentences in the revised manuscript to compare with the interesting study of Schiferl et al., (2016): In section 3.1.2: "In addition, inter-annual variabilities of NH3 concentrations over the United-States are dominated by meteorological conditions [Schiferl et al., 2016]." In section 3.2.1: "This is a different finding than in Schiferl et al. (2016) since they restricted IASI high relative errors when comparing to the GEOS-Chem model over the United-States, which inherently favors larger columns and thus lead to weaken the observed seasonal cycle."

Referee: In Section 2.2, it is important that the authors report what proportion of the column observations from each satellite were below the limit of detection and how those data were incorporated into the monthly means used throughout the paper. If observations below the limit of detection were discarded, then the resulting monthly means will be biased high. It would then be important to filter the model output in a similar way to ensure that the observation-model comparison is more appropriate. Authors: As mentioned in the manuscript, IASI's detection limit is 4-6 1015 molecules/cm2. Observations below this detection limits represent about 60% of the 2014-2015 dataset. Those were not discarded when computing monthly means. CrIS's detection limit is 1-2 1015 molecules/cm2 but no observations in the current product are reported (Shephard et al., 2019). This is a potential reason why CrIS is high compared to IASI in absolute values (See figure R1). However, when comparing to the model data, we selected CHIMERE outputs located within the same 0.15°x0.15° grid box than the satellite and within 1 hour from its measurement to ensure that the comparisons are appropriate. We now have added a sentence about this difference in averaging IASI and CrIS when comparing monthly means to the model outputs in section 3.2.1: "Note that values below detection limits have not been filtered out from the IASI dataset whereas the quality flag was used to discard CrIS's retrievals associated with DOFS<=0.1 (Section 2.2.2) favors larger observed columns. Consequently, the normalized seasonal cycle

[Figure]

amplitude derived from CrIS is weaker than the IASI one."

Figure R1: Time series of daily mean NH3 concentrations (in molecules/cm2) derived from IASI and CrIS satellite measurements (red and black, respectively), and from the CHIMERE model outputs coincident in space and time with IASI (in blue) and CrIS (in cyan).

Referee: A general concern in Section 3.1 is the confidence with which the authors interpret the causes contributing to the seasonality and IAV of the ammonia columns. In many cases, the explanations provided by the authors seem reasonable, but unless there is conclusive proof, the language should be toned down to indicate that these are possible/likely explanations rather than the only ones: Referee: Lines 282-301, a handful of data are provided to describe farming practices in different regions, but not in a consistent way. What evidence is there that the factors described are the most important in causing the spatial and temporal patterns observed? Authors: We have changed the tone of the text, it is now: "The observed seasonality is mainly related to agricultural practices (fertilizer application period varying as function of the crop types and type of livestock) and changes in temperatures, with higher temperatures favoring volatilization. This likely explains the high concentration in July and August."

Referee: Lines 313-314 How do crop type and phenological stage impact ammonia concentrations leading to interannual variability? Authors: The phenological stage controls the fertilizer spreading dates, driving NH3 emissions, and consequently, is likely to regulate NH3 Inter-annual variability observed in a specific region. We have added details in the manuscript: "It has been recently shown that spatial variability of NH3 emissions in France is due to fertilizer use and type and pedoclimatic conditions, and that temporal variability depends on seasonal timing of fertilizer applications [Ramanantenasoa et al., 2018]. In addition, inter-annual variabilities of NH3 concentrations over the United-States are dominated by meteorological conditions [Schiferl et al., 2016]. Thus, inter-annual variability of observed NH3 total columns is likely to be driven by meteorological conditions and specific agricultural constrains (crop type and

phenological stage for instance).”

Referee: Lines 330-333 These seem like plausible explanation for the impact of precipitation amount of ammonia columns, but is there direct evidence that they are the only (most) important factors? Authors: We added likely and toned down our language throughout this section and in the conclusion.

Referee: Lines 334-335 The relationship between gas phase ammonia and temperature should be exponential based on the temperature dependence of its volatilization (either vapor pressure or effective solubility). Does the correlation coefficient change if a non-linear fit is tried? Authors: We have checked and found a correlation of R = 0.30 instead of 0.33 when using a linear fit. We have rectified the manuscript accordingly. The residuals of the fit, however are similar when trying linear and exponential based fitting.

Referee: In Section 3.2, the authors compare 'standardized' monthly means for the years 2014 and 2015 between the two satellite products and the model. More explanation should be provided about how these standardized means were calculated. Do the emissions used in the model differ between the two years? This would be useful to know to help in interpreting the variability produced by the model. Authors: We have included the computation equations regarding the standardization in the 2.4 section: "The standardized columns have been computed following equation 1: $X\_stand\hat{}data= ((X\hat{}data-\mu(X\hat{}data) ))/(S(X\hat{}data))$ (1) Where $(X\hat{}data )= 1/N \sum\_(i = 1)^N X\_data, S(X\hat{}data) = \sqrt{(1/(N-1) \sum\_(N = 1)^N (X\_i - \mu)ãĂŮ^2 )}$, X\hat{}data corresponds to NH3 columns derived from a dataset (IASI, CrIS, or CHIMERE), and X_stand\hat{}data is the corresponding standardized dataset. " The emissions of the model are the same for the 2 years of simulations; the interannual variability of the model is therefore likely to be attributed to meteorological conditions changes. We have clarified in the text that the emissions were the same for the two years and have added a sentence: "In addition, year-to-year variability can be seen in the model with lower concentrations in March 2015 compared to 2014 for instance, despite constant emissions in the 2-years simulation. This interannual variability is likely to be attributed to meteorological conditions changes."

Referee: Lines 371-382 This discussion is a bit confusing because initially the values quoted from the correlation plots of are the coefficients of determination, and then the comparison is restricted to select months and the values quoted are the slopes. I would recommend quoting the r2 values for both, to make it more clear that the coefficients of determination did not increase significantly when the months were restricted. Also, the fact that the slope is close to 1 is not that meaningful since each dataset has already been standardized. Authors: We have changed the text accordingly by removing the slope values and adding p-value instead: "Over the whole period, the coefficient of determination (r2) between the standardized monthly mean NH3 columns derived from IASI (CrIS), and the CHIMERE model is 0.58 (0.18) for the annual cycles of 2014 and 2015 with low associated p-values of 1.5 10-5 (0.06) reflecting the significance level of the fits (not shown here). If we only consider months of high NH3 in the domain from March to August, the correlation between the observational datasets and the model is rather good with r2 values between IASI (CrIS) and CHIMERE of 0.29 (0.14) with associated p-values of 0.07 (0.24), as shown in Figure 7. Since annual total emissions are the same for the two years and simply disaggregated with a monthly profile in the model, the correlations reveal that the seasonal cycle is likely to be reproduced by the model. In addition, year-to-year variability can be seen in the model with lower concentrations in March 2015 compared to 2014 for instance, despite constant emissions in the 2-years simulation. This interannual variability is likely to be attributed to meteorological conditions changes. However, the values of the r2 lower than 0.5 indicate that the CHIMERE model only reproduces at most half of the observed monthly temporal NH3 variabilities in the domain. Similar variabilities are found between the observations and the model outputs since the coefficients of correlation of the standard deviations are 0.4 and 0.6 between CHIMERE and IASI and CrIS, respectively." We have also changed the abstract accordingly: "A detailed analysis of the seasonal cycle is performed using both IASI and the CrIS instrument data, together with outputs from the CHIMERE atmospheric model. For 2014 and 2015 the CHIMERE model shows coefficient of determination of 0.58 and 0.18 when comparing with IASI and CrIS, respectively."

Referee: In Section 3.3, which focuses on the role of NH3 in producing PM2.5 in the Ile de France region, the analysis is overly simplistic. Why have the PM2.5 observations included in the analysis been restricted the measurements between 9 and 11 am? This time interval is particularly challenging to interpret because of the impacts of primary emissions and the role of the rapidly changing boundary layer height. It seems like a poor choice of time window to focus on a phenomenon that is influenced by long-range/regional transport of a precursor species like NH3. The role of temperature and relative humidity on the formation of ammonium salts is well-described by thermodynamic relationships. Statements like those on Lines 504-509 are not fully accurate. Authors: Over the studied area, Metop-A and Metop-B have an overpass time difference ranging from only a few seconds to 67 minutes depending on the viewing geometry of the satellite scans; the average difference is of 26 minutes for the 1325 days of common measurements. Over the whole time period IASI (MetopA and B) overpass time is about 9.50am on average. Therefore we have selected PM2.5 data between 9 and 11 am to study cases in which PM2.5 and NH3 (observations averaged with MetopA and B) concentrations are enhanced simultaneously (or within a one-hour interval) over the IdF region. We also tried a similar analysis considering PM2.5 measured at 10am only and averaged all day (between 8am and 6pm), and this did not change our results regarding the number of events detected for case A and B. Concerning the statements concerning the role of temperature and humidity on the formation of ammonium salts, we have added 'mainly' and 'in particular' to be more accurate: "Our observations are in agreement with previous studies [Bessagnet et al., 2016; Wang et al., 2015], which have shown that the formation of ammonium salt needs a specific humidity of 60 - 70%, mainly because it corresponds to the deliquescence point of NH4NO3 in ambient air. This is in agreement with our results since the mean of relative humidity in case A is 70%. Our results also support the idea that a relatively low atmospheric temperature favor PM2.5 formation in particular since the phase equilibrium leads to NH4NO3 decomposition above 30 °C."

Specific comments: Referee: Line 46 – 'biochemical' should perhaps be 'biogeochemical' Authors: We changed this. Referee: Line 63 – 'related to' should be 'relative to' Authors: We changed this. Referee: Line 111-114 – It would be helpful to reword the sentence slightly, to clarify that all of the studies being referenced were carried out in Paris. Authors: We have reworded this sentence as: "However, although the Paris megacity is repeatedly shrouded by particulate pollution episodes, many studies are limited in the Paris megacity and performed over relatively short time frame during field campaigns: NH3 measurements from May 2010 to February 2011 [Petetin et al., 2016] and nitrate, sulfate, and ammonium aerosol measurements in July 2009 [Zhang et al., 2013], or based on numerical simulations [Skyllakou et al., 2014]." Referee: Figure 1 – The coloring of the map by the emissions is not easy to see. The colors become a very different shade on the map then on the legend. Is it possible to use a map that doesn't have a green background, or to make the emissions coloring more opaque? Authors: We changed the background of the map and made the emissions coloring more opaque. Referee: Figure 6 – would be helpful to have the same months identified on the axis for each year Authors: We have edited the figure to have the same months for the 2 years. References: Shephard, M. W., Dammers, E., Kharol, S., and Cady-Pereira, K.: Ammonia measurements from space with the Cross-track Infrared Sounder (CrIS): characteristics and applications, in preparation for ACP, 2019

Please also note the supplement to this comment:
https://www.atmos-chem-phys-discuss.net/acp-2019-138/acp-2019-138-AC1-supplement.pdf
* * *
[Figure]

[Figure]

**Fig. 1.**

[Figure]

---

## Author Comment (AC2) · 4 Sep 2019

"Atmospheric ammonia variability and link with PM formation: a case study over the Paris area" by Camille Viatte et al. Anonymous Referee #2

Referee: In this study, Viatte et al. use satellite observations (CrIS, IASI) to a) characterize the spatial and inter annual variability of ammonia column over Western Europe and its drivers and b) examine the connection between NH3 and PM2.5 over Paris. The material presented is interesting and well suited for ACP However, I have some significant concerns regarding the robustness of some of the conclusions and the lack of connection between a) and b). These need to be addressed before I publication can be considered. Authors: We would like to thank the referee for his/her insightful com-

ments. We have performed additional analyses and adapted the manuscript to fully address those comments.

General Comments Referee: a) there are places when the authors make fairly definitive claims with insufficient support/references. For instance Line 49: it is stated that N causes species/ecosystem extinction. A specific reference is needed. Authors: We have added 2 references for this sentence: [Isbell et al., 2013; Hernandez et al., 2016]

Referee: Line 341 and discussion above. This discussion is too speculative and needs to be much better supported. Was more corn planted in 2011 than in 2012? Were planting dates shifted earlier in 2011 relative to 2012? This is critical since the authors then state that they have shown that meteorology and farming practices account for the interannual variability in NH3 column. Authors: We have toned down our language to indicate that these are possible/likely explanations rather than the only ones.

Referee: Line 374 It is stated that the correlation is "good" based on Fig. 7 (r2<0.3) . What is the p value, what is the uncertainty on the slopes given the large error bars shows in Fig. 7? In general, the authors need to be more quantitative when reporting statistics: always give p value for correlation (e.g ., line 331 and 333) and uncertainty for slopes. Authors: We have changed "good" to "rather good". As proposed by the other referee, the values of the slopes are not that meaningful since each dataset has already been standardized. Therefore we have removed the slope values and added the p-values for each r2 values, as you suggested. "Over the whole period, the co-efficient of determination (r2) between the standardized monthly mean NH3 columns derived from IASI (CrIS), and the CHIMERE model is 0.58 (0.18) for the annual cycles of 2014 and 2015 with low associated p-values of 1.5 10-5 (0.06) reflecting the signif-icance level of the fits (not shown here). If we only consider months of high NH3 in the domain from March to August, the correlation between the observational datasets and the model is rather good with r2 values between IASI (CrIS) and CHIMERE of 0.29 (0.14) with associated p-values of 0.07 (0.24), as shown in Figure 7. Since annual total emissions are the same for the two years and simply disaggregated with a monthly

profile in the model, the correlations reveal that the seasonal cycle is likely to be reproduced by the model. In addition, year-to-year variability can be seen in the model with lower concentrations in March 2015 compared to 2014 for instance, despite constant emissions in the 2-years simulation. This interannual variability is likely to be attributed to meteorological conditions changes. However, the values of the r2 lower than 0.5 indicate that the CHIMERE model only reproduces at most half of the observed monthly temporal NH3 variabilities in the domain. Similar variabilities are found between the observations and the model outputs since the coefficients of correlation of the standard deviations are 0.4 and 0.6 between CHIMERE and IASI and CrIS, respectively." We have also changed the abstract accordingly: "A detailed analysis of the seasonal cycle is performed using both IASI and the CrIS instrument data, together with outputs from the CHIMERE atmospheric model. For 2014 and 2015 the CHIMERE model shows coefficient of determination of 0.58 and 0.18 when comparing with IASI and CrIS, respectively."

Referee: b) there is very little connection between a) and b) in the current manuscript. In part b), the authors focus on the relationship between PM2.5 and NH3 in two (fairly similar) years (2014, 2015). The main conclusion is that meteorology (temperature, local PBL) probably controls whether NH3 contributes to PM2.5. This is interesting although very much expected from studies performed in other regions. From part a), I was instead expecting the authors to consider whether the considerable variability in NH3 sources over Belgium/Netherlands could impact PM2.5 over Paris. From part a), I was also expecting to have the authors show whether CHIMERE is able to capture the observed correlation between PM2.5 and NH3. This could help understand whether the observed PM2.5 enhancement results from production of ammonium nitrate in Ile de France or from transport of ammonium nitrate/sulfate or other aerosols from Belgium. I fully appreciate that such analysis will require significant work. However, without a significantly stronger connection between part a) and b), I would recommend the paper be split, with part a) being more readily publishable. Authors: We have added a section (3.3) and a Figure (new Figure 11) to evaluate the capacity of the model to

reproduce PM2.5 over the Parisian region. "Comparisons of PM2.5 concentrations in IdF derived from the Airparif network and CHIMERE for 2014 and 2015 To evaluate the model capacity to reproduce PM2.5 concentrations over the Parisian region, comparisons between the Airparif measurements network and the CHIMERE outputs have been performed for 2014 and 2015 (Figure 11). For those years, concentrations of PM2.5 are measured hourly from the surface at 13 Airparif stations distributed over the IdF region (black dots, Figure 1). To compare with the CHIMERE model, we have extracted the hourly surface PM2.5 outputs in the IdF region, i. e. within a 50 km-radius circle from Paris. Results of the comparison are shown in Figure 11. Day-to-Day variability of PM2.5 concentrations at the surface is well represented by the CHIMERE model with however differences during pollution events in March/April and in December for both years. The model may underestimate PM2.5 concentrations in spring due to unknown PM2.5 formation processes, but overestimate them in winter which could be due to uncertainties on NH3 emissions from wood burning processes. Overall, good agreement is found between the measurements and the model in term of PM2.5 concentrations over the IdF region given values of r2 of 0.56 (associated with p-value of 6 10-133), a slope of 0.67 $\pm$ 3.51, with a slightly underestimation of the CHIMERE model given a mean relative difference (calculated as model-observations/observations) of -18% over 2014 and 2015." We have also added a sentence in the conclusion about this analysis: "In this region, we also found that the CHIMERE model is able to reproduce the day-to-day variability of PM2.5 concentrations (r2 of 0.56), with however an underestimation during spring pollution events, which could be due to unknown secondary aerosol formation processes." Finally, we have added a sentence in the abstract section about PM2.5 concentrations evaluation from CHIMERE: "In addition, PM2.5 concentrations derived from the CHIMERE model have been evaluated against surface measurements from the Airparif network over Paris. Agreement was found (r2 of 0.56) with however an underestimation during spring pollution events." To investigate whether the variability in NH3 sources over the northeast part of the domain could impact NH3 over Paris, we have studied the cross-correlation function of NH3 concentrations between

[Figure]

the Northeast part of the domain (over the Netherlands) and the IdF region (see Figure R1 and Figure S5 in the supplement information). The cross-correlation function (CCF) is calculated between the daily averaged mean of the IASI NH3 columns observed over these two regions (both are average values of available pixels of the same day). From the CCF plot, we can see that when lag = 0 (i.e. within the same day), the cross-correlation is maximum with CCF = 0.37, and the CCF is above 0.3 when lag=±1 (i.e. 1 day before or after) for the whole time period (2008-2016). Therefore, correlation between NH3 concentrations over the northeast part of the domain and the IdF region is relatively correlated. This confirms the result suggested by the back-trajectory analysis in Figure 10. We have also computed the CCF over these two regions considering months with high NH3: the maximum CCF between March and August and between March and April are 0.35 and 0.26, respectively. Therefore we have added a sentence about this analysis in the new section 3.4: "Indeed, NH3 columns over the Netherlands are relatively correlated to NH3 columns measured over IdF since the cross-correlation function is 0.37 at lag = 0 and above 0.3 at lag = ±1 day over the whole time period (2008-2016 - Figure S5)." and we add a sentence in the abstract : "Variability of NH3 in the Northeast region is likely to impact NH3 concentrations in the Parisian region since the cross-correlation function is above 0.3 (at lag = 0 and 1). "

Figure R1: Cross-correlation analysis of NH3 concentrations between the Northeast part of the domain (over the Netherlands) and the IdF region. In addition, to study the effect of transport on NH3 and PM2.5 concentrations observed over the Parisian region, we have included wind fields analysis in Section 3.4 (old Section 3.3). In Figure 12 (old Figure 11) in the lower panel, we have added wind fields parameters (direction and speed) from ERA-5 and included wind roses for studies cases (ensemble, case A, and case B) in the supplement information. Results of the statistic show that cases involving simultaneous enhancements of NH3 and PM2.5 concentrations in Paris (cases A) are associated with wind fields dominantly coming from the Northeast. Airmasses coming from this area are thus likely to favor simultaneous enhancements of NH3 and PM2.5 over Paris. We have added few sentences in the new Section 3.4 and

the conclusion about this: Section 3.4: "Results also suggest that simultaneous enhancements of NH3 and PM2.5 over Paris (cases A) are mainly associated with wind fields dominantly coming from the Northeast part of the domain (Figure S6). Thus the combination of the following four meteorological parameters favors simultaneous appearances of NH3 and of PM2.5 in Paris (i.e. case A): low surface temperatures (5°C), with thin boundary layers (∼500m), rare precipitations, and northeast wind." In the conclusion section: "To assess the link between NH3 and PM2.5 over the Parisian (IdF) region, the main meteorological parameters driving the optimal conditions involved in the PM2.5 formation have been identified. The results show that relatively low temperature, thin boundary layer, coupled with almost no precipitation and wind coming from the northeast, favor the PM2.5 formation with the presence of atmospheric NH3 in the IdF region."

Technical comments Referee: Section 2.3 the description of CHIMERE is far too short (especially with respect to the treatment of ammonia. For instance: -> how is dry deposition represented? Does it include the bidirectional exchange between land and atmosphere -> what is the temporal resolution of the emissions? Does it include a diurnal cycle? It would be useful to show the seasonality of the emissions in a few regions, to help the reader better analyze Figs 2 and 3 -> how is the gas/aerosol partitioning of NH3 represented (ISORROPIA?) -> I assume that NH3/NH4/NH4NO3 in CHIMERE have been evaluated previously? Please provide reference for these studies at this stage. I also encourage the authors to show how the configuration of CHIMERE that is used here performs against surface observations (e.g., EMEP wet deposition/concentrations). This could be briefly discussed in the main text, with figures in the supplementary materials. Authors: We have detailed the description of the model by adding this section: "These annual emissions are then distributed in hourly data to feed CHIMERE using seasonal, weekly and hourly factors. Fire emissions come from the Global Fire Assimilation System (GFAS, [Kaiser et al., 2012]). The model computes hourly concentrations for more than 180 species, among which are the regulated pollutants such as ozone, PM10, and NH3. The processes that will influence the NH3

concentrations taken into consideration in CHIMERE are the dry deposition (following [Wesely et al., 1989] and wet deposition due to in-cloud process and precipitations. The gas-particulate phase equilibrium is computed with the ISOROPPIA module [Nenes et al, 1998] which is a thermodynamic equilibrium model for NH4+, NO3- and SO42-. It evaluates the NH4NO3 contribution to the particulate matter which is especially large during March-April pollution episodes [Petit et al., 2017]."

Referee: Section 3.1.1 It would be useful to include a map showing the distribution of livestock and major crops in Western Europe so that the reader can see the relationship between NH3 emissions and the different sources described by the authors. This would be especially helpful as some of the material the authors refer to is in French. Authors: We have added specific references for livestock mapping and found English versions of the references: • https://agriculture.gouv.fr/overview-french-agricultural-diversity ; • Scarlat et al., 2018 – their figure 2], • [Robinson et al., 2014 - their figure 2c].

Referee: Fig. 5. This figures shows first and foremost that there is good correlation between skin temperature and precipitation at the regional level. I think it would be more relevant to show the relationship between temperature/precipitation and NH3 anomaly. In addition, I assume that the precipitation/temperature anomalies exhibit some significant spatial variability? Do you weigh the anomaly by the average NH3 column? High NH3 columns only cover a small fraction of your domain and it's unclear to me why it would respond to the average temperature change (vs the local change). Authors: We have tried the analysis suggested by the referee. Anomalies of NH3 and temperature/precipitation over the domain are shown in Figure R2. The results suggests strong relationships exists between anomalies of NH3 and skin temperature (correlation R = 0.72), and total precipitation (anti-correlation R = -52).

Figure R2: monthly mean anomaly (relative to the 10-years – 2008 to 2017 - monthly average) of total precipitation/skin temperature derived from ECMWF from March to August in the domain, versus NH3 total columns anomaly derived from IASI. When computing the anomalies, temperature and precipitation anomalies were not weighting

by NH3 total column.

Referee: Section 3.2. I am a little confused by the need for the standardization. CrIS and IASI seem reasonably close, so why not use the model absolute NH3 column. In addition, Fig. 6 only show one CHIMERE time series, shouldn't there be two, one for CHIMERE sampled at the IASI overpass time and one at the CrIS overpass time (with AK).. Authors: The CrIS and the IASI data are not close in absolute values: CrIS is higher than IASI in the region of interest (of about 1.1016 molecule/cm2). In addition, the CHIMERE output concentrations are closer to IASI observations than CrIS's ones (see Figure R3), which is why we wanted to standardized each dataset independently. We have also tested the comparison between CrIS and CHIMERE by taking into account the different vertical sensitivity (smoothing by the AK) but results were not improved.

Figure R3: Time series of dailymean NH3 concentrations (in molecules/cm2) derived from IASI and CrIS satellite measurements (red and black, respectively), and from the CHIMERE model outputs coincident in space and time with IASI (in blue) and CrIS (in cyan). As for Figure 6, we have changed it to include the CHIMERE time series sampled in space and time with IASI and CrIS, as you suggested.

Referee: Line 351 I am not sure I understand the motivation for picking this years. Why not use the climatological seasonality? Why are these years more useful to benchmark the model? They look fairly similar as far as I can tell from the supporting material. Authors: In the frame of evaluating the model capacity of reproducing NH3 variability in space and time at regional scale and its impact on air quality at local scale, those two years are interesting for the following reasons. At regional scale (over the 400 km radius around Paris), NH3 total columns derived from IASI in 2014 and 2015 are highly variable in time throughout the years and especially in spring, reaching 10% higher in March and 50% lower in May than the 10-years average. Since ammonia emission variability depends on seasonal timing of fertilizer applications in France [Ramanantenasoa et al., 2018], this period is crucial to assess the model capacity. Second, for

[Figure]

those two years NH3 concentrations over the IdF region (100 km radius around Paris) are also extremely high in March (Figure R4, upper panel). These extreme events might have affected the Parisian air quality since PM2.5 concentrations are also enhanced, especially in 2014 (Figure R4, lower panel). We have added this Figure in the Supplementary Information (Figure S1). Therefore, we think these years could serve as benchmark to evaluate the model in terms of NH3 variability at regional scale, and PM2.5 formation at local scale. We have changed the manuscript to explain the motivation for choosing these years in section 2.3 dedicated to the CHIMERE model: "To evaluate the model capacity of reproducing NH3 variability in space and time at regional scale and its impact on air quality at local scale, comparisons have been performed in 2014 and 2015 for the following reasons. At regional scale (over the 400 km radius around Paris), NH3 total columns derived from IASI in 2014 and 2015 are highly variable in spring, reaching 10% higher in March and 50% lower in May than the 10-years average. Since ammonia emission variability in France depends on seasonal timing of fertilizer applications [Ramanantenasoa et al., 2018], this period is crucial to assess the model capacity. Second, the IdF region (100 km radius around Paris) also experiences high NH3 and PM2.5 events in spring 2014 and 2015 (Figure S1). Thus, these years serve as benchmark to evaluate the model in terms of NH3 variability and PM2.5 formation at local and regional scales."

Figure R4: Time series of daily mean NH3 concentrations (in molecules/cm2) derived from IASI (upper panel) and PM2.5 concentration (in in $\mu$g/m3) observed over the IdF region between 2013 and 2016.

Technical comments Referee: They are a few issues with language. It sometimes (rarely) makes it challenging to understand the manuscript. Referee: line 28: regression slope. Remove slope Authors: We have removed slope Referee: line 63: related->relative Authors: We have changed this. Referee: Line 112: many of studies? Authors: We have deleted "of" Referee: Line 283: farming species? Do you mean livestock? Authors: Yes, we have changed it to livestock. Referee: Line 300. What are

non-poultry granivorous (animals)? Authors: We have deleted granivorous. Referee: Fig. 7 What do the error bars correspond to? Authors: The error bars correspond to the 1-sigma standard deviation around the mean. We have clarified it in the figure caption. Referee: Fig. 9: Same than Fig.7 -> "Same as Fig. 8" Authors: We have changed this. Referee: Fig. 12: Define IQR Authors: We added: The IQR is the "interquartile range", and it equals to Q3 - Q1 where Q3 and Q1 are the 75th and 25th percentiles. Setting the thresholds at Q1 - 1.5 * IQR and Q3 + 1.5 * IQR is a common practice to determine outliers. Referee: Line 220: I don't understand the distinction between inorganic, organic and natural aerosols? Authors: We have deleted this part of the text to include more specific description of the model. Referee: Line 487. Why is the value given on line 476 different (mean/median?) Authors: The first value refers to the example given in the manuscript, i. e. from March 3rd and March 19th 2014, whereas the second value represents the mean value for the case A over the whole dataset. We have added 'over the whole dataset' in the latest sentence to avoid confusion.

Please also note the supplement to this comment:
https://www.atmos-chem-phys-discuss.net/acp-2019-138/acp-2019-138-AC2-supplement.pdf

———————————————————

[Figure]

[Figure]

**Fig. 1.**

[Figure]

Fig. 2.

[Figure]

[Figure]

Fig. 3.

Over IdF

IASI NH$_3$ total column

NH$_3$ colonne ($10^{16}$ mole/cm$^2$)

Legend: 2013, 2014, 2015, 2016

Airparif PM$_{2.5}$

PM$_{2.5}$ (µg/m$^3$)

Legend: 2013, 2014, 2015, 2016

Time (Feb, Mar, Apr, May, Jun)

**Fig. 4.**

---

## Author Response (AR1)

**"Atmospheric ammonia variability and link with PM formation: a case study over the Paris area" by Camille Viatte et al.**

**Anonymous Referee #1**

Authors: We would like to thank the referee for his/her insightful comments. We have made changes to the manuscript to address those comments.

Referee: This manuscript uses observations from two independent satellites to assess the role of $NH_3$ in springtime particle pollution episodes in the Paris region by examining the seasonal and interannual variability (IAV) in $NH_3$ columns over northwestern Europe.

The observations are compared to simulations from the CHIMERE chemical transport model. In general, the authors do a good job of reviewing the existing literature to provide context for their results, but it would be useful if they could include a comparison with the study of Schiferl et al. (2016), which examines seasonal cycles and IAV of $NH_3$ over the US.

Authors: We have added sentences in the revised manuscript to compare with the interesting study of Schiferl et al., (2016):

In section 3.1.2: "In addition, inter-annual variabilities of $NH_3$ concentrations over the United-States are dominated by meteorological conditions [Schiferl et al., 2016]."

In section 3.2.1: "This is a different finding than in Schiferl et al. (2016) since they restricted IASI high relative errors when comparing to the GEOS-Chem model over the United-States, which inherently favors larger columns and thus lead to weaken the observed seasonal cycle."

Referee: In Section 2.2, it is important that the authors report what proportion of the column observations from each satellite were below the limit of detection and how those data were incorporated into the monthly means used throughout the paper. If observations below the limit of detection were discarded, then the resulting monthly means will be biased high. It would then be important to filter the model output in a similar way to ensure that the observation-model comparison is more appropriate.

Authors: As mentioned in the manuscript, IASI's detection limit is 4-6 $10^{15}$ molecules/cm$^2$. Observations below this detection limits represent about 60% of the 2014-2015 dataset. Those were not discarded when computing monthly means. CrIS's detection limit is 1-2 $10^{15}$ molecules/cm$^2$ but no observations in the current product are reported (Shephard et al., 2019). This is a potential reason why CrIS is high compared to IASI in absolute values (See figure R1). However, when comparing to the model data, we selected CHIMERE outputs located within the same 0.15°x0.15° grid box than the satellite and within 1 hour from its measurement to ensure that the comparisons are appropriate.

We now have added a sentence about this difference in averaging IASI and CrIS when comparing monthly means to the model outputs in section 3.2.1: "Note that values below detection limits have not been filtered out from the IASI dataset whereas the quality flag was used to discard CrIS's retrievals associated with DOFS<=0.1 (Section 2.2.2) favors larger observed columns. Consequently, the normalized seasonal cycle amplitude derived from CrIS is weaker than the IASI one."

[Figure]

*Figure R1: Time series of daily mean NH₃ concentrations (in molecules/cm²) derived from IASI and CrIS satellite measurements (red and black, respectively), and from the CHIMERE model outputs coincident in space and time with IASI (in blue) and CrIS (in cyan).*

Referee: A general concern in Section 3.1 is the confidence with which the authors interpret the causes contributing to the seasonality and IAV of the ammonia columns. In many cases, the explanations provided by the authors seem reasonable, but unless there is conclusive proof, the language should be toned down to indicate that these are possible/likely explanations rather than the only ones:

Referee: Lines 282-301, a handful of data are provided to describe farming practices in different regions, but not in a consistent way. What evidence is there that the factors described are the most important in causing the spatial and temporal patterns observed?

Authors: We have changed the tone of the text, it is now: "The observed seasonality is mainly related to agricultural practices (fertilizer application period varying as function of the crop types and type of livestock) and changes in temperatures, with higher temperatures favoring volatilization. This likely explains the high concentration in July and August."

Referee: Lines 313-314 How do crop type and phenological stage impact ammonia concentrations leading to interannual variability?

Authors: The phenological stage controls the fertilizer spreading dates, driving $NH_3$ emissions, and consequently, is likely to regulate $NH_3$ Inter-annual variability observed in a specific region.

We have added details in the manuscript: "It has been recently shown that spatial variability of $NH_3$ emissions in France is due to fertilizer use and type and pedoclimatic conditions, and that temporal variability depends on seasonal timing of fertilizer applications [Ramanantenasoa et al., 2018]. In addition, inter-annual variabilities of $NH_3$ concentrations over the United-States are dominated by meteorological conditions [Schiferl et al., 2016]. Thus, inter-annual variability of observed $NH_3$ total columns is likely to be driven by meteorological conditions and specific agricultural constrains (crop type and phenological stage for instance)."

Referee: Lines 330-333 These seem like plausible explanation for the impact of precipitation amount of ammonia columns, but is there direct evidence that they are the only (most) important factors?

Authors: We added likely and toned down our language throughout this section and in the conclusion.

Referee: Lines 334-335 The relationship between gas phase ammonia and temperature should be exponential based on the temperature dependence of its volatilization (either vapor pressure or effective solubility). Does the correlation coefficient change if a non-linear fit is tried?

Authors: We have checked and found a correlation of R = 0.30 instead of 0.33 when using a linear fit. We have rectified the manuscript accordingly. The residuals of the fit, however are similar when trying linear and exponential based fitting.

Referee: In Section 3.2, the authors compare 'standardized' monthly means for the years 2014 and 2015 between the two satellite products and the model. More explanation should be provided about how these standardized means were calculated. Do the emissions used in the model differ between the two years? This would be useful to know to help in interpreting the variability produced by the model.

Authors: We have included the computation equations regarding the standardization in the 2.4 section: "The standardized columns have been computed following equation 1:

$$X_{stand}^{data} = \frac{(X^{data} - \mu(X^{data}))}{S(X^{data})} \quad (1)$$

Where $(X^{data}) = \frac{1}{N}\sum_{i=1}^{N} X_i^{data}$ , $S(X^{data}) = \sqrt{\frac{1}{N-1}\sum_{N=1}^{N}(X_i - \mu)^2}$, $X^{data}$ corresponds to $NH_3$ columns derived from a dataset (IASI, CrIS, or CHIMERE), and $X_{stand}^{data}$ is the corresponding standardized dataset. "

The emissions of the model are the same for the 2 years of simulations; the interannual variability of the model is therefore likely to be attributed to meteorological conditions changes. We have clarified in the text that the emissions were the same for the two years and have added a sentence: "In addition, year-to-year variability can be seen in the model with lower concentrations in March 2015 compared to 2014 for instance, despite constant emissions in the 2-years simulation. This interannual variability is likely to be attributed to meteorological conditions changes."

Referee: Lines 371-382 This discussion is a bit confusing because initially the values quoted from the correlation plots of are the coefficients of determination, and then the comparison is restricted to select months and the values quoted are the slopes. I would recommend quoting the r2 values for both, to make it more clear that the coefficients of determination did not increase significantly when the months were restricted. Also, the fact that the slope is close to 1 is not that meaningful since each dataset has already been standardized.

Authors: We have changed the text accordingly by removing the slope values and adding p-value instead:

"Over the whole period, the coefficient of determination ($r^2$) between the standardized monthly mean $NH_3$ columns derived from IASI (CrIS), and the CHIMERE model is 0.58 (0.18) for the annual cycles of 2014 and 2015 with low associated p-values of 1.5 $10^{-5}$ (0.06) reflecting the significance level of the fits (not shown here). If we only consider months of high $NH_3$ in the domain from March to August, the correlation between the observational datasets and the model is rather good with $r^2$ values between IASI (CrIS) and CHIMERE of 0.29 (0.14) with associated p-values of 0.07 (0.24), as shown in Figure 7. Since annual total emissions are the same for the two years and simply disaggregated with a monthly profile in the model, the correlations reveal that the seasonal cycle is likely to be reproduced by the model. In addition, year-to-year variability can be seen in the model with lower concentrations in March 2015 compared to 2014 for instance, despite constant emissions in the 2-years simulation. This interannual variability is likely to be attributed to meteorological conditions changes. However, the values of the $r^2$ lower than 0.5 indicate that the CHIMERE model only reproduces at most half of the observed monthly temporal $NH_3$ variabilities in the domain. Similar variabilities are found between the observations and the model outputs since the coefficients of correlation of the standard deviations are 0.4 and 0.6 between CHIMERE and IASI and CrIS, respectively."

We have also changed the abstract accordingly:

"A detailed analysis of the seasonal cycle is performed using both IASI and the CrIS instrument data, together with outputs from the CHIMERE atmospheric model. For 2014 and 2015 the CHIMERE model shows coefficient of determination of 0.58 and 0.18 when comparing with IASI and CrIS, respectively."

Referee: In Section 3.3, which focuses on the role of $NH_3$ in producing $PM_{2.5}$ in the Ile de France region, the analysis is overly simplistic. Why have the $PM_{2.5}$ observations included in the analysis been restricted the measurements between 9 and 11 am? This time interval is particularly challenging to interpret because of the impacts of primary emissions and the role of the rapidly changing boundary layer height. It seems like a poor choice of time window to focus on a phenomenon that is influenced by long-range/regional transport of a precursor species like NH3. The role of temperature and relative humidity on the formation of ammonium salts is well-described by thermodynamic relationships. Statements like those on Lines 504-509 are not fully accurate.

Authors: Over the studied area, Metop-A and Metop-B have an overpass time difference ranging from only a few seconds to 67 minutes depending on the viewing geometry of the satellite scans; the average difference is of 26 minutes for the 1325 days of common measurements. Over the whole time period IASI (MetopA and B) overpass time is about 9.50am on average. Therefore we have selected $PM_{2.5}$ data between 9 and 11 am to study cases in which $PM_{2.5}$ and $NH_3$ (observations averaged with MetopA and B) concentrations are enhanced simultaneously (or within a one-hour interval) over the IdF region. We also tried a similar analysis considering $PM_{2.5}$ measured at 10am only and averaged all day (between 8am and 6pm), and this did not change our results regarding the number of events detected for case A and B.

Concerning the statements concerning the role of temperature and humidity on the formation of ammonium salts, we have added 'mainly' and 'in particular' to be more accurate: "Our observations are in agreement with previous studies [Bessagnet et al., 2016; Wang et al., 2015], which have shown that the formation of ammonium salt needs a specific humidity of 60 - 70%, mainly because it corresponds to the deliquescence point of $NH_4NO_3$ in ambient air. This is in agreement with our results since the mean of relative humidity in case A is 70%. Our results also support the idea that a relatively low atmospheric temperature favor $PM_{2.5}$ formation in particular since the phase equilibrium leads to $NH_4NO_3$ decomposition above 30 °C."

Specific comments:

Referee: Line 46 – 'biochemical' should perhaps be 'biogeochemical'

Authors: We changed this.

Referee: Line 63 – 'related to' should be 'relative to'

Authors: We changed this.

Referee: Line 111-114 – It would be helpful to reword the sentence slightly, to clarify that all of the studies being referenced were carried out in Paris.

Authors: We have reworded this sentence as: "However, although the Paris megacity is repeatedly shrouded by particulate pollution episodes, many studies are limited in the Paris megacity and performed over relatively short time frame during field campaigns: $NH_3$ measurements from May 2010 to February 2011 [Petetin et al., 2016] and nitrate, sulfate, and ammonium aerosol measurements in July 2009 [Zhang et al., 2013], or based on numerical simulations [Skyllakou et al., 2014]."

Referee: Figure 1 – The coloring of the map by the emissions is not easy to see. The colors become a very different shade on the map then on the legend. Is it possible to use a map that doesn't have a green background, or to make the emissions coloring more opaque?

Authors: We changed the background of the map and made the emissions coloring more opaque.

Referee: Figure 6 – would be helpful to have the same months identified on the axis for each year

Authors: We have edited the figure to have the same months for the 2 years.

References: Shephard, M. W., Dammers, E., Kharol, S., and Cady-Pereira, K.: Ammonia measurements from space with the Cross-track Infrared Sounder (CrIS): characteristics and applications, in preparation for ACP, 2019

**"Atmospheric ammonia variability and link with PM formation: a case study over the Paris area" by Camille Viatte et al.**

**Anonymous Referee #2**

Referee: In this study, Viatte et al. use satellite observations (CrIS, IASI) to a) characterize the spatial and inter annual variability of ammonia column over Western Europe and its drivers and b) examine the connection between NH3 and PM2.5 over Paris. The material presented is interesting and well suited for ACP However, I have some significant concerns regarding the robustness of some of the conclusions and the lack of connection between a) and b). These need to be addressed before I publication can be considered.

Authors: We would like to thank the referee for his/her insightful comments. We have performed additional analyses and adapted the manuscript to fully address those comments.

**General Comments**

Referee: a) there are places when the authors make fairly definitive claims with insufficient support/references.

For instance Line 49: it is stated that N causes species/ecosystem extinction. A specific reference is needed.

Authors: We have added 2 references for this sentence: [Isbell et al., 2013; Hernandez et al., 2016]

Referee: Line 341 and discussion above. This discussion is too speculative and needs to be much better supported. Was more corn planted in 2011 than in 2012? Were planting dates shifted earlier in 2011 relative to 2012? This is critical since the authors then state that they have shown that meteorology and farming practices account for the interannual variability in NH3 column.

Authors: We have toned down our language to indicate that these are possible/likely explanations rather than the only ones.

Referee: Line 374 It is stated that the correlation is "good" based on Fig. 7 (r2<0.3) . What is the p value, what is the uncertainty on the slopes given the large error bars shows in Fig. 7? In general, the authors need to be more quantitative when reporting statistics: always give p value for correlation (e.g ., line 331 and 333) and uncertainty for slopes.

Authors: We have changed "good" to "rather good". As proposed by the other referee, the values of the slopes are not that meaningful since each dataset has already been standardized. Therefore we have removed the slope values and added the p-values for each $r^2$ values, as you suggested.

"Over the whole period, the coefficient of determination ($r^2$) between the standardized monthly mean $NH_3$ columns derived from IASI (CrIS), and the CHIMERE model is 0.58 (0.18) for the annual cycles of 2014 and 2015 with low associated p-values of $1.5 \cdot 10^{-5}$ (0.06) reflecting the significance level of the fits (not shown here). If we only consider months of high $NH_3$ in the domain from March to August, the correlation between the observational datasets and the model is rather good with $r^2$ values between IASI (CrIS) and CHIMERE of 0.29 (0.14) with associated p-values of 0.07 (0.24), as shown in Figure 7. Since annual total emissions are the same for the two years and simply disaggregated with a monthly profile in the model, the correlations reveal that the seasonal cycle is likely to be reproduced by the model. In addition, year-to-year variability can be seen in the model with lower concentrations in March 2015 compared to 2014 for instance, despite constant emissions in the 2-years simulation. This interannual variability is likely to be attributed to meteorological conditions changes. However, the values of the $r^2$ lower than 0.5 indicate that the CHIMERE model only reproduces at most half of the observed monthly temporal $NH_3$ variabilities in the domain. Similar variabilities are found between the observations and the model outputs since the coefficients of correlation of the standard deviations are 0.4 and 0.6 between CHIMERE and IASI and CrIS, respectively."

We have also changed the abstract accordingly:

"A detailed analysis of the seasonal cycle is performed using both IASI and the CrIS instrument data, together with outputs from the CHIMERE atmospheric model. For 2014 and 2015 the CHIMERE model shows coefficient of determination of 0.58 and 0.18 when comparing with IASI and CrIS, respectively."

Referee: b) there is very little connection between a) and b) in the current manuscript. In part b), the authors focus on the relationship between PM2.5 and NH3 in two (fairly similar) years (2014, 2015). The main conclusion is that meteorology (temperature, local PBL) probably controls whether NH3 contributes to PM2.5. This is interesting although very much expected from studies performed in other regions. From part a), I was instead expecting the authors to consider whether the considerable variability in NH3 sources over Belgium/Netherlands could impact PM2.5 over Paris. From part a), I was also expecting to have the authors show whether CHIMERE is able to capture the observed correlation between PM2.5 and NH3. This could help understand whether the observed PM2.5 enhancement results from production of ammonium nitrate in Ile de France or from transport of ammonium nitrate/sulfate or other aerosols from Belgium. I fully appreciate that such analysis will require significant work. However, without a significantly stronger connection between part a) and b), I would recommend the paper be split, with part a) being more readily publishable.

Authors: We have added a section (3.3) and a Figure (new Figure 11) to evaluate the capacity of the model to reproduce PM$_{2.5}$ over the Parisian region.

"Comparisons of PM$_{2.5}$ concentrations in IdF derived from the Airparif network and CHIMERE for 2014 and 2015

To evaluate the model capacity to reproduce PM$_{2.5}$ concentrations over the Parisian region, comparisons between the Airparif measurements network and the CHIMERE outputs have been performed for 2014 and 2015 (Figure 11). For those years, concentrations of PM$_{2.5}$ are measured hourly from the surface at 13 Airparif stations distributed over the IdF region (black dots, Figure 1). To compare with the CHIMERE model, we have extracted the hourly surface PM$_{2.5}$ outputs in the IdF region, i. e. within a 50 km-radius circle from Paris.

Results of the comparison are shown in Figure 11. Day-to-Day variability of PM$_{2.5}$ concentrations at the surface is well represented by the CHIMERE model with however differences during pollution events in March/April and in December for both years. The model may underestimate PM$_{2.5}$ concentrations in spring due to unknown PM$_{2.5}$ formation processes, but overestimate them in winter which could be due to uncertainties on NH$_3$ emissions from wood burning processes. Overall, good agreement is found between the measurements and the model in term of PM$_{2.5}$ concentrations over the IdF region given values of r$^2$ of 0.56 (associated with p-value of 6 10$^{-133}$), a slope of 0.67 ± 3.51, with a slightly underestimation of the CHIMERE model given a mean relative difference (calculated as model-observations/observations) of -18% over 2014 and 2015."

We have also added a sentence in the conclusion about this analysis: "In this region, we also found that the CHIMERE model is able to reproduce the day-to-day variability of PM$_{2.5}$ concentrations (r$^2$ of 0.56), with however an underestimation during spring pollution events, which could be due to unknown secondary aerosol formation processes."

Finally, we have added a sentence in the abstract section about PM$_{2.5}$ concentrations evaluation from CHIMERE: "In addition, PM$_{2.5}$ concentrations derived from the CHIMERE model have been evaluated against surface measurements from the Airparif network over Paris. Agreement was found (r$^2$ of 0.56) with however an underestimation during spring pollution events."

To investigate whether the variability in NH$_3$ sources over the northeast part of the domain could impact NH$_3$ over Paris, we have studied the cross-correlation function of NH$_3$ concentrations between the Northeast part of the domain (over the Netherlands) and the IdF region (see Figure R1 and Figure S5 in the supplement information). The cross-correlation function (CCF) is calculated between the daily averaged mean of the IASI NH$_3$ columns observed over these two regions (both are average values of available pixels of the same day). From the CCF plot, we can see that when lag = 0 (i.e. within the same day), the cross-correlation is maximum with CCF = 0.37, and the CCF is above 0.3 when lag=±1 (i.e. 1 day before or after) for the whole time period (2008-2016). Therefore, correlation between NH$_3$ concentrations over the northeast part of the domain and the IdF region is relatively correlated. This confirms the result suggested by the back-trajectory analysis in Figure 10. We have also computed the CCF over these two regions considering months with high NH$_3$: the maximum CCF between March and August and between March and April are 0.35 and 0.26, respectively. Therefore we have added a sentence about this analysis in the new section 3.4: "Indeed, NH$_3$ columns over the Netherlands are relatively correlated to NH$_3$ columns measured over IdF since the cross-correlation function is 0.37 at lag

= 0 and above 0.3 at lag = ±1 day over the whole time period (2008-2016 - Figure S5)." and we add a sentence in the abstract : "Variability of $NH_3$ in the Northeast region is likely to impact $NH_3$ concentrations in the Parisian region since the cross-correlation function is above 0.3 (at lag = 0 and 1). "

[Figure]

*Figure R1: Cross-correlation analysis of $NH_3$ concentrations between the Northeast part of the domain (over the Netherlands) and the IdF region.*

In addition, to study the effect of transport on $NH_3$ and $PM_{2.5}$ concentrations observed over the Parisian region, we have included wind fields analysis in Section 3.4 (old Section 3.3). In Figure 12 (old Figure 11) in the lower panel, we have added wind fields parameters (direction and speed) from ERA-5 and included wind roses for studies cases (ensemble, case A, and case B) in the supplement information. Results of the statistic show that cases involving simultaneous enhancements of $NH_3$ and $PM_{2.5}$ concentrations in Paris (cases A) are associated with wind fields dominantly coming from the Northeast. Airmasses coming from this area are thus likely to favor simultaneous enhancements of $NH_3$ and $PM_{2.5}$ over Paris.  We have added few sentences in the new Section 3.4 and the conclusion about this:  Section 3.4: "Results also suggest that simultaneous enhancements of $NH_3$ and $PM_{2.5}$ over Paris (cases A) are mainly associated with wind fields dominantly coming from the Northeast part of the domain (Figure S6). Thus the combination of the following four meteorological parameters favors simultaneous appearances of $NH_3$ and of $PM_{2.5}$ in Paris (i.e. case A): low surface temperatures (5°C), with thin boundary layers (~500m), rare precipitations, and northeast wind." In the conclusion section: "To assess the link between $NH_3$ and $PM_{2.5}$ over the Parisian (IdF) region, the main meteorological parameters driving the optimal conditions involved in the $PM_{2.5}$ formation have been identified. The results show that relatively low temperature, thin boundary layer, coupled with almost no precipitation and wind coming from the northeast, favor the $PM_{2.5}$ formation with the presence of atmospheric $NH_3$ in the IdF region."

Referee: Section 2.3 the description of CHIMERE is far too short (especially with respect to the treatment of ammonia. For instance: -> how is dry deposition represented? Does it include the bidirectional exchange between land and atmosphere -> what is the temporal resolution of the emissions? Does it include a diurnal cycle? It would be useful to show the seasonality of the emissions in a few regions, to help the reader better analyze Figs 2 and 3 -> how is the gas/aerosol partitioning of NH3 represented (ISORROPIA?) -> I assume that NH3/NH4/NH4NO3 in CHIMERE have been evaluated previously? Please provide reference for these studies at this stage. I also encourage the authors to show how the configuration of CHIMERE that is used here performs against surface observations (e.g., EMEP wet deposition/concentrations). This could be briefly discussed in the main text, with figures in the supplementary materials.

Authors: We have detailed the description of the model by adding this section:

"These annual emissions are then distributed in hourly data to feed CHIMERE using seasonal, weekly and hourly factors. Fire emissions come from the Global Fire Assimilation System (GFAS, [Kaiser et al., 2012]).

The model computes hourly concentrations for more than 180 species, among which are the regulated pollutants such as ozone, $PM_{10}$, and $NH_3$. The processes that will influence the $NH_3$ concentrations taken into consideration in CHIMERE are the dry deposition (following [Wesely et al., 1989] and wet deposition due to in-cloud process and precipitations. The gas-particulate phase equilibrium is computed with the ISOROPPIA module [Nenes et al, 1998] which is a thermodynamic equilibrium model for $NH_4^+$, $NO_3^-$ and $SO_4^{2-}$. It evaluates the $NH_4NO_3$ contribution to the particulate matter which is especially large during March-April pollution episodes [Petit et al., 2017]."

Referee: Section 3.1.1 It would be useful to include a map showing the distribution of livestock and major crops in Western Europe so that the reader can see the relationship between NH3 emissions and the different sources described by the authors. This would be especially helpful as some of the material the authors refer to is in French.

Authors: We have added specific references for livestock mapping and found English versions of the references:

- https://agriculture.gouv.fr/overview-french-agricultural-diversity ;
- Scarlat et al., 2018 – their figure 2],
- [Robinson et al., 2014 - their figure 2c].

Referee: Fig. 5. This figures shows first and foremost that there is good correlation between skin temperature and precipitation at the regional level. I think it would be more relevant to show the relationship between temperature/precipitation and NH3 anomaly. In addition, I assume that the precipitation/temperature anomalies exhibit some significant spatial variability? Do you weigh the anomaly by the average $NH_3$ column? High $NH_3$ columns only cover a small fraction of your domain and it's unclear to me why it would respond to the average temperature change (vs the local change).

Authors: We have tried the analysis suggested by the referee. Anomalies of $NH_3$ and temperature/precipitation over the domain are shown in Figure R2. The results suggests strong relationships exists between anomalies of $NH_3$ and skin temperature (correlation R = 0.72), and total precipitation (anti-correlation R = -52).

[Figure]

*Figure R2: monthly mean anomaly (relative to the 10-years – 2008 to 2017 - monthly average) of total precipitation/skin temperature derived from ECMWF from March to August in the domain, versus $NH_3$ total columns anomaly derived from IASI.*

When computing the anomalies, temperature and precipitation anomalies were not weighting by $NH_3$ total column.

Referee: Section 3.2. I am a little confused by the need for the standardization. CrIS and IASI seem reasonably close, so why not use the model absolute NH3 column. In addition, Fig. 6 only show one CHIMERE time series, shouldn't there be two, one for CHIMERE sampled at the IASI overpass time and one at the CrIS overpass time (with AK)..

Authors: The CrIS and the IASI data are not close in absolute values: CrIS is higher than IASI in the region of interest (of about $1.10^{16}$ molecule/cm$^2$). In addition, the CHIMERE output concentrations are closer to

IASI observations than CrIS's ones (see Figure R3), which is why we wanted to standardized each dataset independently. We have also tested the comparison between CrIS and CHIMERE by taking into account the different vertical sensitivity (smoothing by the AK) but results were not improved.

[Figure]

*Figure R3: Time series of dailymean NH₃ concentrations (in molecules/cm²) derived from IASI and CrIS satellite measurements (red and black, respectively), and from the CHIMERE model outputs coincident in space and time with IASI (in blue) and CrIS (in cyan).*

As for Figure 6, we have changed it to include the CHIMERE time series sampled in space and time with IASI and CrIS, as you suggested.

Referee: Line 351 I am not sure I understand the motivation for picking this years. Why not use the climatological seasonality? Why are these years more useful to benchmark the model? They look fairly similar as far as I can tell from the supporting material.

Authors: In the frame of evaluating the model capacity of reproducing NH₃ variability in space and time at regional scale and its impact on air quality at local scale, those two years are interesting for the following reasons.

At regional scale (over the 400 km radius around Paris), NH₃ total columns derived from IASI in 2014 and 2015 are highly variable in time throughout the years and especially in spring, reaching 10% higher in March and 50% lower in May than the 10-years average. Since ammonia emission variability depends on seasonal timing of fertilizer applications in France [Ramanantenasoa et al., 2018], this period is crucial to assess the model capacity.

Second, for those two years NH₃ concentrations over the IdF region (100 km radius around Paris) are also extremely high in March (Figure R4, upper panel). These extreme events might have affected the Parisian air quality since PM$_{2.5}$ concentrations are also enhanced, especially in 2014 (Figure R4, lower panel). We have added this Figure in the Supplementary Information (Figure S1).

Therefore, we think these years could serve as benchmark to evaluate the model in terms of NH$_3$ variability at regional scale, and PM$_{2.5}$ formation at local scale. We have changed the manuscript to explain the motivation for choosing these years in section 2.3 dedicated to the CHIMERE model: "To evaluate the model capacity of reproducing NH$_3$ variability in space and time at regional scale and its impact on air quality at local scale, comparisons have been performed in 2014 and 2015 for the following reasons. At regional scale (over the 400 km radius around Paris), NH$_3$ total columns derived from IASI in 2014 and 2015 are highly variable in spring, reaching 10% higher in March and 50% lower in May than the 10-years average. Since ammonia emission variability in France depends on seasonal timing of fertilizer applications [Ramanantenasoa et al., 2018], this period is crucial to assess the model capacity. Second, the IdF region (100 km radius around Paris) also experiences high NH$_3$ and PM$_{2.5}$ events in spring 2014 and 2015 (Figure S1). Thus, these years serve as benchmark to evaluate the model in terms of NH$_3$ variability and PM$_{2.5}$ formation at local and regional scales."

[Figure]

*Figure R4: Time series of daily mean NH$_3$ concentrations (in molecules/cm$^2$) derived from IASI (upper panel) and PM$_{2.5}$ concentration (in in µg/m$^3$) observed over the IdF region between 2013 and 2016.*

Referee: They are a few issues with language. It sometimes (rarely) makes it challenging to understand the manuscript.

Referee: line 28: regression slope. Remove slope

Authors: We have removed slope

Referee: line 63: related->relative

Authors: We have changed this.

Referee: Line 112: many of studies?

Authors: We have deleted "of"

Referee: Line 283: farming species? Do you mean livestock?

Authors: Yes, we have changed it to livestock.

Referee: Line 300. What are non-poultry granivorous (animals)?

Authors: We have deleted granivorous.

Referee: Fig. 7 What do the error bars correspond to?

Authors: The error bars correspond to the 1-sigma standard deviation around the mean. We have clarified it in the figure caption.

Referee: Fig. 9: Same than Fig.7 -> "Same as Fig. 8"

Authors: We have changed this.

Referee: Fig. 12: Define IQR

Authors: We added: The IQR is the "interquartile range", and it equals to Q3 - Q1 where Q3 and Q1 are the $75t^h$ and $25^{th}$ percentiles. Setting the thresholds at Q1 - 1.5 * IQR and Q3 + 1.5 * IQR is a common practice to determine outliers.

Referee: Line 220: I don't understand the distinction between inorganic, organic and natural aerosols?
Authors: We have deleted this part of the text to include more specific description of the model.

Referee: Line 487. Why is the value given on line 476 different (mean/median?)

Authors: The first value refers to the example given in the manuscript, i. e. from March $3^{rd}$ and March $19^{th}$ 2014, whereas the second value represents the mean value for the case A over the whole dataset. We have added 'over the whole dataset' in the latest sentence to avoid confusion.

[revised manuscript text omitted]

---

## Author Response (AR2)

**Comments to Editor**

Authors: We would like to thank the editor for his insightful comments. We have made changes to the manuscript to address those comments.

Editor: Line 13-46: Abstract is too long, it would be good if you can shorten this.
Authors: We have deleted few sentences to shorten the abstract.

Editor: Line 21: variability
Author: we changed it

Editor: Line 34: agreement on what?
Authors: we changed it to "In addition, $PM_{2.5}$ concentrations derived from the CHIMERE model have been evaluated against surface measurements from the Airparif network over Paris, for which agreement was found ($r^2$ of 0.56) with however an underestimation during spring pollution events. "

Editor: Line 38: the importance of long-range transport
Authors: we modified this.

Editor: Line 42: use spring instead of springtime
Authors: we changed it

Editor: Line 44: factors such as
Author: It has been modified

Editor: Line 29: In terms of, in the case of, with respect to, Regarding are the alternatives of this " In term of"
Authors: we changed to "With respect to"

Editor: Line 64: protocol on reduction of ammonia? If yes, then please specify that.
Authors: we specified this in the sentence.

Editor: Line 178: mean relative difference will not show the seasonal or any other bias in the measurements
Authors: we agree so we have added this to the sentence: "and an underestimation by IASI ranging from 10 to 50 %"

Editor: Line 196-199: I do not understand this sentence. Please rephrase
Authors: This sentence has been rephrased such as: 'To account for any a priori information used in the retrieval (i.e. observation operator) in air quality model comparisons and data assimilation into models, the CRPR provides the retrieved error covariance and averaging kernels.'

Line 223-224: the sentence is not complete
Authors: We have rephrased this sentence: "The model computes hourly concentrations for more than 180 species, including the regulated pollutants such as ozone, PM10, and NH3."

Editor: Line 226: use "precipitation" and at other places
Author: It has been changed throughout the manuscript

Editor: Line 234: "This makes a reasonable assumption …"
Author: we changed the sentence accordingly

Editor: Line 298, line 539: Use "Note that" or "It should be noted that" something like this
Author: We modified the sentence

Editor: Line 314: main region of mineral.
Author: We changed it

Editor: Line 348: monthly mean anomaly is calculated with respect to the ten-year average data
Author: We changed the sentence accordingly

Editor: Line 353: To further examine the analyses,
Author: we changed this sentence

Editor: Line 367: What is fertilizer spreading?
Authors: we have changed this sentence to 'Springtime is a fertilizer application period'

Editor: Line 368: when the temperatures are relatively lower as in the case of 2012
Authors: It has been changed

Editor: Line 374: factors account for the higher NH3… (not parameters, but the factors or processes)
Author: We have changed it to "Overall, our results suggest that variability in meteorological (precipitation and temperature) and farming practices (fertilizer and manure applications) may play an important role in driving the large inter-annual variability in $NH_3$ column observed by IASI and CrIS in the domain of study."

Editor: Line 413: model results is
Authors: Changed to "If we only consider months of high $NH_3$ in the domain from March to August, the correlation between the observational datasets and the model results is weaker with $r^2$ values between IASI (CrIS) and CHIMERE of 0.29 (0.14) and is not significant (p>0.1) against CrIS, as shown in Figure 7."

Editor: Line 431: "First one can note that"? Write something like "It can be noted that" or "Note that."
Authors: we changed to It can be noted that

Editor: Line 471: outputs are
Author: Changed to "outputs in"

Editor: Line 481-485: difficult to understand, please rephrase
Authors: We have rephrased to "To investigate the impact of intensive agriculture practices on the Paris megacity air quality, we need to better understand the role of NH3 in the formation of PM2.5. This process depends, among others, on specific meteorological conditions such as atmospheric temperature and humidity that alter the gas-particle partitioning."

Editor: Line 483: "slight underestimation"
Author: we modified this sentence

Editor: Line 494: Using the 10-years of
Author: Changed

Editor: Line 531: difficult to quantify the "enough" here
Authors: We have modified to "This time period was selected to have the most IASI observations (combining Metop-A and B) in the IdF region."

Editor: Line 534: and case B
Author: Changed

Editor: Line 536: 3 March, 19 March like that (in lines 540, 543, 544 too)
Authors: we changed all the dates.

Editor: Line 546: a detailed analysis is made for the study period (not for the whole data sets)
Authors: We changed it.

Editor: Line 556: What are rare precipitations?
Author: We have changed it to 'almost no precipitation'

Editor: Line 557: northeast wind.
Authors: we have modified the text

**Comments to referee 2**

Authors: We would like to thank the referee 2 for his insightful comments. We have made changes to the manuscript to address those comments.
Referee: line 41
when air masses are originated -> when air masses originate
Authors: We have changed this.

Referee: line 44
specify unit for the lag
Authors: we added 'day'

Referee: line 94
underestimates the NH3 budget -> do you mean the NH3 (surface?) concentration
We have changed the sentence to "CHIMERE model underestimates the $NH_3$ surface concentrations and emissions over Paris [Petetin et al., 2016; Fortems-Cheiney et al., 2016]".

Referee: line 120
limited in -> limited to
Authors: We have changed this.

Referee: line 233
It evaluates -> it simulates?
Authors: We have changed this.

Referee: There are other instances where sentences can be difficult to understand. Careful proofreading is needed.
Authors: we have carefully proofread the revised manuscript.

Referee: Line 372, 375. Please provide p values. Relatively->also
Authors: We have added p values.

Referee: Line 378. Please rephrase sentence.
Authors: we have changed it to "However, the values of the r2 lower than 0.6 (0.2) indicate that the CHIMERE model only reproduces at most half (20%) of the monthly temporal NH3 variabilities observed by IASI (CrIS) in the domain."

Referee: Line 385. I would suggest to rephrase along these lines.
Overall, our results suggests that variability in meteorological and farming practices may play an important role in driving the large inter-annual variability in NH3 column observed by IASI and CrIS.
We have changed these lines accordingly.

Referee: line 422 with low associated p-values of 1.5 10-5 (0.06) reflecting the significance level of the fits (not shown here) -> (p<0.1)
Authors: we have changed this.

Referee: line 425-426 rather good is not quantitative. Instead, you may want to convey that the correlation is weaker (0.29 and 0.14, respectively) and is not significant (p<0.1) against CrIS

Authors: we changed it to : "If we only consider months of high $NH_3$ in the domain from March to August, the correlation between the observational datasets and the model results is weaker with $r^2$ values between IASI (CrIS) and CHIMERE of 0.29 (0.14) and is not significant (p>0.1) against CrIS, as shown in Figure 7."

Referee: line 432. r2=0.18 against CrIS. So it only captures ~20% of the monthly variability.

Authors : we changed it to "However, the values of the r2 lower than 0.6 (0.2) indicate that the CHIMERE model only reproduces at most half (20%) of the monthly temporal NH3 variabilities observed by IASI (CrIS) in the domain."

line 434. I don't understand what the authors mean here. What is the correlation between standard deviations (which is a scalar)?

Authors : This sentence is confusing, we choose to delete it.

Referee: line 495. associated with p-value of 6x10-133. -> it's probably enough to just write (p<0.05)

Authors: We changed this.

Referee: line 497. I assume that you are referring to the mean normalized bias?

Author: Absolutely, we changed the name accordingly.

Referee: line 526. Relatively correlated is not quantitative. If the correlation is significant (at p<0.05 or p<0.1), just state it.

Authors: We have changed the sentence. Since the number of observations for the CCF plot over the whole tome period (2008-2016) is large (N = 2735), we assumed a normal distribution of CCF with a mean of 0 and a standard deviation of 0.0382. So the upper limit for the 95% interval level is fixed at 0.03748. Since all the CCF values in the plot (-15 < lag < 15) are above 0.03748, they are all significant at the 95% confidential level. However, this is not a hypothesis test, so we cannot calculate the p-value.

Referee: line 527. I am puzzled with the +/- lag. Given the hypothesis that NH3 originates from Northern Europe, why consider +lag?

Authors: To support the idea that NH3 originates from Northern Europe, and not the contrary, we have to compare the value of cross-correlation function (CCF) for both lag > 0 (i.e. enhancements of NH3 concentrations observed over Northern Europe followed by enhancements over IdF) and lag < 0 (i.e. enhancements of NH3 concentrations over Northern Europe preceded by enhancements over IdF).

Referee: line 562. Which test did the author use to establish the significance? The mean of B falls within 1sigma of the mean of A.

Authors: We have performed the Wilcoxon-Mann-Whitney test for which values where significant (p<0.01). We have noted the p value in the text.

Referee: Fig. 13. I suggest you reiterate the definition of A and B in the caption and refer to Fig. 12. Also indicate the number of observations used for each case (N ensemble?)

Authors: we have changed the caption accordingly.